# Forecasting green roof detention performance by temporal downscaling of precipitation time-series projections

Vincent Pons[1,2], Rasmus Benestad[3], Edvard Siversten[4], Tone Merete Muthanna[1], and Jean-Luc Bertrand-Krajewski[2]

[1]Department of Civil and Environmental Engineering, The Norwegian University of Science and Technology, Trondheim, 7031, Norway
[2]Univ Lyon, INSA Lyon, DEEP, EA7429, 11 rue de la Physique, F-69621, Villeurbanne cedex, France
[3]Norwegian Meteorological Institute, Oslo, Norway
[4]SINTEF AS, S.P. Andersens veg 3, N-7465 Trondheim, Norway

**Correspondence:** Vincent Pons (vincent.pons@ntnu.no)

**Abstract.**

A strategy to evaluate the suitability of different Multiplicative-Random-Cascades to produce rainfall time-series, taking into account climate change, inputs for green infrastructures models. The Multiplicative-Random-Cascades reproduce a (multi)fractal distribution of precipitation through an iterative and multiplicative random process. In the current study, the initial model, a flexible cascade that deviates from multifractal scale invariance, was improved with *i)* a temperature dependency and *ii)* an additional function to reproduce the temporal structure of rainfall. The structure of the models with depth and temperature dependency was found to be applicable in eight locations studied across Norway and France. The resulting time-series from both reference period and projection based on RCP 8.5 were applied to two green roofs with different properties. The different models led to a slight change in the performance of green roofs, but this was not significant compared to the range of outcomes due to ensemble uncertainty in climate modelling and the stochastic uncertainty due to the nature of the process. The hydrological dampening effect of the green infrastructure was found to decrease in most of the Norwegian cities due to an increase in precipitation, especially Bergen (Norway), while slightly increasing in Marseille (France) due to decrease in rainfall events frequency.

## 1 Introduction

Hydrologic performance of stormwater Green Infrastructure ($GI$) is usually divided between retention and detention. Retention refers to water stored, infiltrated, or evapotranspired. Actual evapotranspiration can be estimated from a water balance including Potential Evapotranspiration, accumulated precipitation, a soil moisture evaluation function and, a crop factor (Johannessen et al., 2017; Oudin et al., 2005). The temporal resolution for modelled evapotranspiration process for green infrastructure is typically daily (Stovin et al., 2013) or hourly (Kristvik et al., 2019). Detention refers to water temporarily stored in the $GI$ before being discharged into a downstream stormwater network. The process temporal resolution is typically minutes. Consequently, modelling $GI$ detention performance requires higher resolution data to estimate its outflow (Schilling, 1991). Therefore, both

**Table 1.** Table of abbreviations

| Abbreviation | Meaning |
| --- | --- |
| $GI$ | Green Infrastructures |
| $MRC$ | Multiplicative Random Cascades |
| $IDF$ curves | Intensity-Duration-Frequency curves |
| $NVE$ | Norwegian Water Resources and Energy Directorate |
| $MET$ | Norwegian Meteorological institute |
| $PET$ | Potential EvapoTranspiration |
| $AET$ | Actual EvapoTranspiration |
| E-Green roof | Extensive green roof |
| D-Green roof | Detention-based extensive green roof |
| RCP 8.5 | Representative Concentration Pathway scenario with a $8.5 W/m^2$ radiative forcing in 2100 |
| $NSE$ | Nash-Sutcliffe Efficiency |
| $VM$ | Variational Method |
| $S_{i,2j}$ | Rainfall continuity indicator at time-step $i$ and time-scale $2j$ |
| $d_{i,2j}$ | Depth [mm] at time-step $i$ and time-scale $2j$ |
| $w_{i,2j}$ | Minimum weight at time-step $i$ and time-scale $2j$ from aggregation of depths $D_{2i,j}$ and $D_{2i+1,j}$ |
| $S_W$ | Random discrete variable of neighbour of the highest weight |
| $CF$ | Climate Factor |

high resolution climate data and projections at sub daily and sub hourly scales are needed in order to model $GI$, and to estimate their potential as a climate change adaptation measure.

In Norway and most of the European countries, precipitation has been measured with tipping buckets in numerous cities from 25 years to decades. Moreover, climate projection at daily resolution for future precipitation and temperature from the EURO-CORDEX project are available at $1 \times 1$ km spatial resolution in Norway (Dyrrdal et al., 2018) and $12 \times 12$ km resolution in France (Jacob et al., 2014). Consequently, the use of such data by urban hydrologists to assess the resilience of $GI$ solutions to face climate change is conditioned by the possibility to downscale them to a sub-hourly resolution.

Downscaling includes two families of methods: Dynamical downscaling and Statistical downscaling (Benestad, 2016). Dy-30 namical downscaling methods use physically based equations, and are usually computationally expensive specially to obtain high resolution data. Statistical downscaling consists in improving the resolution of data based on statistical properties observed from a lower resolution dataset. The computational cost is lower, therefore, statistical methods might still be used to fill the gap in the next decades until the computational power is sufficient. In addition, the use of stochastic approach is necessary

due to the current limitation in parametrization of small scale processes (below the truncation scale) and the lack of coupling between resolved and parametrized scales (Sanchez et al., 2016).

Statistical downscaling has already been extensively used to temporally downscale data for various temporal resolutions, usually hourly or daily data. Three popular methods can be mentioned: *i)* the method of fragment, *ii)* the method based on point process theory, and *iii)* the method of multiplicative random cascades. The method of fragment (Li et al., 2018; Lu et al., 2015) is a resampling method based on k-nearest neighbours (Kalra and Ahmad, 2011), which has been applied to derive hourly data from daily data. It can be accurate and effective due to its resampling nature, but it requires a large dataset, and by its design it cannot ensure extrapolation from observed data. Therefore, it might not be suitable to downscale climate projections. Methods based on point process theory have been used (Glasbey et al., 1995; Onof et al., 2000). The main principle is to generate storm occurrences, and then describe them based on rain cells and statistical distribution based on Poisson point process. Multiplicative random cascades ($MRC$) consist of using successively random cascades to split data in N data of finer resolution ($N = 2$ in most of the cases). It is a very popular method that deserves further investigations (Gaur and Lacasse, 2018; Rupp et al., 2012; Thober et al., 2014). They were originally based on the hypothesis of multifractal scale invariance (Schertzer and Lovejoy, 1987) and were further developped by Gupta and Waymire (1993) and Olsson (1998). While multifractal scale invariance remain studied (Gires et al., 2020), several studies noted a deviation from that behaviour which led to the use of more flexible models (Koutsoyiannis and Langousis, 2011; Veneziano et al., 2006). Multiplicative random cascades can be divided between canonical and micro-canonical types. The canonical one ensures conservation on average while the micro-canonical one ensures exact conservation. The parameters of the canonical $MRC$ are often calibrated by fitting between observed and simulated non-centred moments of depths or intensity through the time-scale (Paschalis et al., 2012). The principle of micro-canonical $MRC$ is usually based on reverse cascades: studying how the data are split and then reproducing the properties of the weights distribution depending on different quantities. The influences of time-scale, rainfall intensity (Paschalis et al., 2012; Rupp et al., 2009) or season (McIntyre et al., 2016) have been extensively studied. Lombardo et al. (2012) suggested that the commonly used $MRC$ suffers from conceptual weaknesses due to the non-stationary process of autocorrelation and proposed a method to improve the model. More recently, (Bürger et al., 2014, 2019) suggested to include a temperature dependency in $MRC$ models to make them more robust. This also enables them to be used with projections.

Green infrastructures, due to their retention and detention capacities, are seen as a promising solution to manage stormwater and cope with climate change, especially in cities where urbanization increases. Among green infrastructures, green roofs are especially suitable for dense urban centers. They are designed to retain day-to-day rain by evapotranspiration and attenuate major rainfall events (Stovin, 2010). Depending on their characteristics they can also help to detain extreme rainfall (Hamouz et al., 2020). Due to the time-scale of their detention process, and their sensitivity to initial water-content at the beginning of a rainfall event, they are suitable for evaluating downscaled time-series. Moreover, it is especially relevant to evaluate their detention performance by the end of the century under a scenario such as RCP 8.5 (Thorndahl and Andersen, 2021). The results could be used to evaluate, at strategical level, their potential in mitigating stormwater in order to make robust decision (Walker et al., 2013).

While downscaling models have been used to model the performance of green infrastructure under current climate (Stovin et al., 2017), or applied to Intensity-Duration-Frequency ($IDF$) curves to do an event based simulation of local stormwater measures (Kristvik et al., 2019), none has been developed to produce future high resolution time-series as input for green infrastructure models. The aim of this research is to evaluate different $MRC$ downscaling models and their potential to produce input time-series to predict the performance of stormwater green infrastructure, for the case of green roofs. In order to achieve this aim, different parts are detailed in the paper: *i)* the development of a general structure of $MRC$; *ii)* the improvement of the $MRC$ structure by adding a temperature dependency , *iii)* the addition of an ordering function to improve the temporal structure of the produced rainfall time-series; *iv)* the evaluation of the capability to reproduce the performance of $GI$ based on observed data; and finally *v)* the analysis of a possible shift in performance of $GI$ at the end of the century.

## 2 Methods

### 2.1 Meteorological data

Time-series of precipitation and temperature from six locations in Norway and two in France, representing four different climates (Table 2) according to the Köppen Geiger classification (Peel et al., 2007), were used to apply the downscaling method. In Norway, the precipitation was measured by 0.2 mm Plumatic Kongsberg tipping rain gauges. The rain gauges were not heated and thus did not operate in cold temperature. They were successively replaced by Lambrecht 1518H3 (measuring tip of 0.1 mm) in the 1990s and 2000s. The stations were operated by the Norwegian Water Resources and Energy Directorate ($NVE$) and the Norwegian Meteorological Institute ($MET$). The data were quality checked by the Norwegian Meteorological institute ($MET$) (Lutz et al., 2020). In Lyon and Marseille, precipitation was measured by 0.2 mm Précis-Mécanique tipping bucket rain gauges. Ten climate projections (temperature and precipitation) on daily resolution with the RCP 8.5 for the period from 2071 to 2099 for Norwegian cities were available online https://nedlasting.nve.no/klimadata/kss (Dyrrdal et al., 2018). For Lyon and Marseille (France), twelve climate projections on daily resolution were available for the same period and RCP (2071 to 2099, RCP 8.5) from http://www.drias-climat.fr/. The RCP 8.5, a scenario with a high gas-emission baseline leading to a radiative forcing of $8.5W/m^2$ in 2100, and the end of the century were chosen to test the methods since it was the scenario and period that deviate the most from the current climate among the available data in both countries. In practice, it is relevant to evaluate $GI$ performance at the end of the century but their design could be based on a different period depending on their lifetime.

### 2.2 Downscaling models and workflow

#### 2.2.1 Data aggregation and processing

The historical data were aggregated two by two from 1-minute resolution (resp. 6-minute) to more than 1-day resolution in order to capture a part of the uncertainty linked to the estimation of the parameter of the models . The aggregation was done for each possible time-steps: all multiples of 2 smaller than 1500 min (as there are 1440 min per day). During the process of

**Table 2.** Locations and input data for current and future climate; The climate column gives the Köppen Geiger classification for climate, Observed days is the number of observed days with data. YearPr is the annual precipitation in mm, YearWt the annual number of wet days (>1mm). YearTe is the mean annual temperature; for these three indicators the $5^{th}$, $50^{th}$ and $95^{th}$ percentiles are displayed

| Location | Observed days | Climate | Latitude | Period | YearPr | YearWt | YearTe |
|---|---|---|---|---|---|---|---|
| Bergen – Sandsli, $MET$ 50480, $NVE$ no. 56.1. | 6150 | Cfb | 60.4 | **Obs:** | $1505, 2081, 2504$ | $153, 189, 218$ | $-2.1, 8.0, 17.8$ |
|  |  |  |  | **RCP 8.5:** | $2240, 3012, 4009$ | $169, 201, 238$ | $2.2, 10.3, 19.3$ |
| Bodø – Skivika, $MET$ 82310, $NVE$ no. 165.11. | 7204 | Dfc | 67.3 | **Obs:** | $643, 991, 1858$ | $114, 152, 266$ | $-3.9, 5.4, 15.4$ |
|  |  |  |  | **RCP 8.5:** | $1150, 1600, 2139$ | $147, 178, 214$ | $-0.3, 8.1, 18.2$ |
| Lyon (France), 6-min time-step | 7671 | Cfb | 45.7 | **Obs:** | $706, 865, 1161$ | $80, 97, 114$ | $0.5, 12.8, 24.3$ |
|  |  |  |  | **RCP 8.5:** | $550, 830, 1187$ | $77, 105, 135$ | $3.9, 15.9, 29.9$ |
| Hamar – Hamar II (Disen), $MET$ 12290 | 4011 | Dfb | 60.8 | **Obs:** | $406, 546, 659$ | $70, 92, 105$ | $-9.8, 5.7, 18.6$ |
|  |  |  |  | **RCP 8.5:** | $508, 689, 861$ | $88, 110, 134$ | $-5.3, 8.1, 21.0$ |
| Kristiansand – Sømskleiva, $MET$ 39150, $NVE$ no. 21.49. | 5219 | Cfb | 58.1 | **Obs:** | $1155, 1512, 1868$ | $120, 137, 161$ | $-0.9, 9.7, 18.6$ |
|  |  |  |  | **RCP 8.5:** | $1258, 1662, 2099$ | $115, 142, 167$ | $0.3, 10.0, 20.5$ |
| Kristiansund – Karihola, $MET$ 64300, $NVE$ no. 110.1. | 8664 | Cfb | 63.1 | **Obs:** | $714, 1094, 2521$ | $131, 167, 226$ | $-0.7, 7.8, 16.2$ |
|  |  |  |  | **RCP 8.5:** | $1440, 2051, 2829$ | $153, 192, 230$ | $1.9, 9.6, 18.0$ |
| Marseille (France), 6-min time-step | 7305 | Csa | 43.3 | **Obs:** | $310, 533, 840$ | $35, 49, 64$ | $4.1, 15.2, 26.2$ |
|  |  |  |  | **RCP 8.5:** | $250, 492, 767$ | $33, 50, 68$ | $7.3, 18.0, 30.6$ |
| Trondheim – Risvollan, $MET$ 68230, $NVE$ 123.38. | 10722 | Dfc | 63.4 | **Obs:** | $669, 965, 1256$ | $117, 150, 173$ | $-6.1, 6.3, 17.9$ |
|  |  |  |  | **RCP 8.5:** | $853, 1176, 1599$ | $133, 163, 196$ | $-1.4, 8.6, 19.2$ |

aggregation, both the weights, Eq. 1, and the rainfall continuity indicator, Eq. 2, measuring the proportion of high weight on the side of the highest neighbouring depth were computed. Given $j \in [1..750]$ a time-scale in minutes, $i \in [0..N_{2j}]$ a time-step with $N_{2j}$ the number of time-steps at scale $2j$, and $d_{i,2j}$ a rainfall depth, the weight $w_{i,2j}$, and the indicator $S_{i,2j}$ of the side of the neighbour were calculated according to:

$$w_{i,2j} = \frac{\min(d_{2i,j}, d_{2i+1,j})}{d_{2i,j} + d_{2i+1,j}} \in [0; 0.5] \tag{1}$$

$$S_{i,2j} = \begin{cases} 0, & \text{if } (d_{i-1,2j} = d_{i+1,2j}) \cup (d_{2i,j} = d_{2i+1,j}) \\ 1, & \text{if } (d_{2i,j} > d_{2i+1,j} \cap d_{i-1,2j} > d_{i+1,2j}) \cup (d_{2i,j} < d_{2i+1,j} \cap d_{i-1,2j} < d_{i+1,2j}) \\ 2, & \text{if } (d_{2i,j} > d_{2i+1,j} \cap d_{i-1,2j} < d_{i+1,2j}) \cup (d_{2i,j} < d_{2i+1,j} \cap d_{i-1,2j} > d_{i+1,2j}) \end{cases} \tag{2}$$

### 2.2.2 Downscaling process

The $MRC$ downscaling process consists of transforming daily rainfall depths to rainfall depths at lower time-scale, e.g. one minute, by means of successive distribution of the depth of a parent time-steps between its two children time-steps. The process is repeated by iteration until the desired time-scale is reached. Figure 1 describes the downscaling process. In practice, the downscaling started at 1440 min (1 day) time-step with 8 iterations to reach a time-step of 5.625 min. The results were interpolated and scaled to a 6-min time-step for comparison with observed data. The final time-step of 6 minutes was chosen based on the resolution of original datasets in Lyon and Marseille. Three steps are necessary to downscale a parent time-step to two children time-steps. The occurrence of a zero-weight, i.e. the probability to assign all the water from the parent time-step to only one of the children time-steps (Figure 1, center left), is tested. This property is especially important and acknowledged by other studies. If a zero-weight does not occur, a non-zero-weight $w_{i,2j} \in ]0,0.5]$ is generated from a probability distribution (Eq. 3b). It distribute the depth from the parent time-step between the two children time-steps, as illustrated in Figure 1, center right. Finally, the weights $w_{i,2j}$ and $1 - w_{i,2j}$ have to be assigned to the children time-steps. The occurrence of $S_W$ (Eq. 4), i.e. allocating the highest weight to the children with the neighbour with the highest depth, is tested (Figure 1, bottom).

$$u_{0,i,2j} \sim \mathcal{U}([0,1]),$$
$$\text{if } u_{0,i,2j} < P(W = 0 | S_{time} = 2j, D = d_{i,2j}, T = T_{i,2j}), \text{ then } w_{i,2j} = 0 \tag{3a}$$
$$\text{else, } w_{i,2j} \sim \mathcal{N}_{[0,\frac{1}{2}]}(\frac{1}{2}, \sigma(S_{time} = 2j)^2) \tag{3b}$$

$$P(S_W = 1), \text{with } S_W \in \{0;1\} \tag{4}$$

### 2.2.3 Downscaling models conceptualization and calibration

Based on the observed data, 6 different $MRC$ models were developed. Different mathematical expressions and probabilistic distributions, detailed in appendix A, where defined to represent equations 3a, 3b and 4, depending on the hypothesis inherent to the later described models (Table 3). The models consist in 3 generators: a zero-weight generator, a non-zero-weight generator and a $Stochastic$ $Element$ $Permutation$ generator ($SEP$ generator). Each of the zero-weight and a non-zero-weight generators (Eq. 3) were considered to vary with time-$Scale$ (indicated with $S$ in the model naming). The letter $I$ in the nomenclature indicate a depth/$Intensity$ for the zero-weight generator (Eq. A2). The $Temperature$ dependency for the zero-weight generator

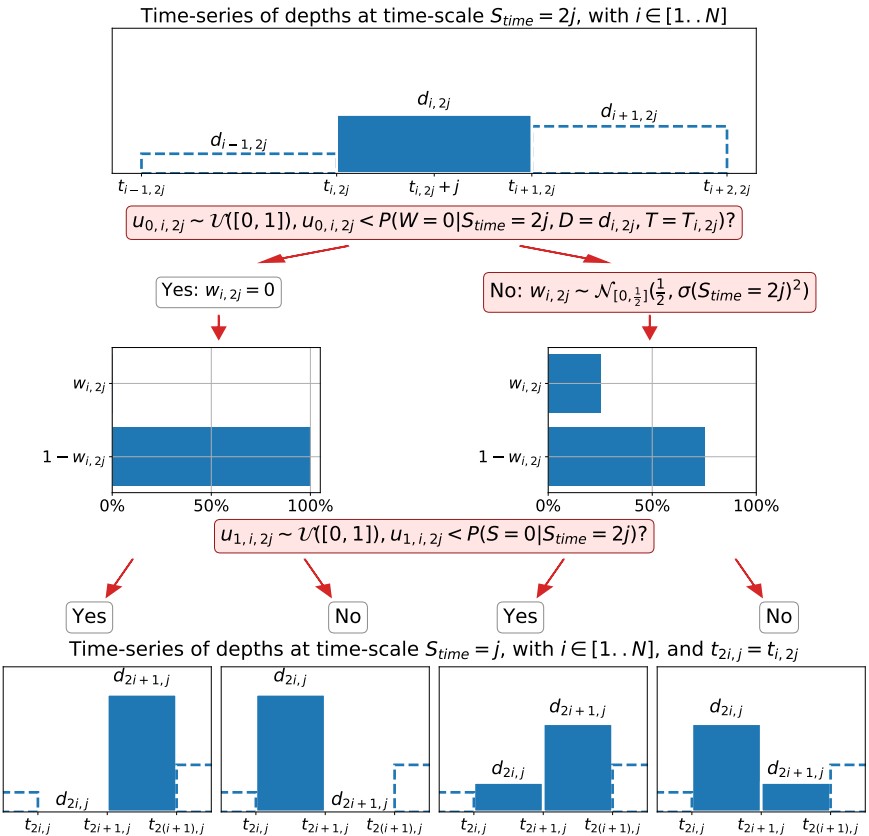

**Figure 1.** Workflow for downscaling to transfer a depth from time-step $T$ to time-step $\frac{T}{2}$. The red boxes involve the generation of a random number. The process starts with 1440 minute time-step to reach 5.625 min an interpolation is then done to reach 6 min time-step.

Eq. A3) was indicated by the letter $T$ in the nomenclature. The temperature dependency was added in an attempt to improve the robustness of the model toward climate change under the hypothesis that the change in rainfall pattern would be correlated to the shift in temperature and that the existing observed datasets already carry the necessary information for calibration. In the models $MRC_{S-SEP}$, $MRC_{SI-SEP}$ and $MRC_{SIT-SEP}$ (Table 3) the weights generated were permuted stochastically depending on the neighbour (indicated with $SEP$, Eq. 4 and A5), while the $MRC_S$, $MRC_{SI}$, and $MRC_{SIT}$ model considered equal probability (0.5) to permute the two children weights.

Similarly to Rupp et al. (2009), the generators of the $MRC$ models include time-scale dependency through analytical formulas. In practice it means that there is a single set of parameters per generator and not a different set at each cascade step. It ensured a relatively parsimonious number of parameters compared to other recent works on microcanonical $MRC$ where a set per cascade step is often used (e.g. 12 to 36 total parameters by Bürger et al. (2019) or from 6 to 224 parameters per cascade step by Müller-Thomy (2020)). It should be noted that based on dataset analysis and as advised by (Serinaldi, 2010), despite the fact that universal and canonical $MRC$ represent the most parsimonious approach parameter wise, their microcanonical

**Table 3.** Nomenclature of the models and various quantities taken into account by each model depending on the process considered; $S$ is the time-scale, $D$ the rainfall depth/intensity, $T$ the temperature and $N$ the close neighbour.

| Model | $P(W=0)$ | | | | $P(W|W \neq 0)$ | | | | $P(S_W=1)$ | | | | Number of parameters |
|---|---|---|---|---|---|---|---|---|---|---|---|---|---|
| | $S$ | $I$ | $T$ | $N$ | $S$ | $I$ | $T$ | $N$ | $S$ | $I$ | $T$ | $N$ | |
| $MRC_S$ | x | | | | x | | | | | | | | 8 |
| $MRC_{S-SEP}$ | x | | | | x | | | | x | | | x | 13 |
| $MRC_{SI}$ | x | x | | | x | | | | | | | | 14 |
| $MRC_{SI-SEP}$ | x | x | | | x | | | | x | | | x | 19 |
| $MRC_{SIT}$ | x | x | x | | x | | | | | | | | 13 |
| $MRC_{SIT-SEP}$ | x | x | x | | x | | | | x | | | x | 18 |

counterpart was preferred. This choice of model that deviate from the hypothesis of multifractal scale invariance was supported by several other studies (Koutsoyiannis and Langousis, 2011; Veneziano et al., 2006). The number of parameter can be lower with the use of universal $MRC$ which were excluded from this study due to lack of flexibility (Serinaldi, 2010). It also allows the model to be used with any desired initial resolution lower than 1500 min. Homogeneity of the resolution in the input datasets was not required for calibration and data processing (i.e. the model can be calibrated using multiple datasets with different resolutions between 1-min and 1 day). The parameters of each generators of $MRC$ models and each locations required calibration. A single-step calibration, based on the processed data, was sufficient for generators with only time-scale dependency. A multiple-steps calibration with data manipulation was necessary for generators with depth/intensity, temperature dependency, and for the non-zero-weight generator. This choice was motivated by the development of the model through data analysis and conceptualization of the model. Especially, the steps and calibration of the 3 generators were chosen to avoid compensation between processes using a bottom-up approach (i.e., starting from local properties and then add dependencies to progressively upscale the model). Additionally, the choice of regression over time-scale was used to avoid parameter sets that lead to the correct distribution of precipitation intensities without temporal consistency. Later studies can further improve the procedure to make it more easily calibrated. The optimizations were based on non-linear least squares the standard library scipy.optimize with default parameters in Python (Virtanen et al., 2020).

- The parameters of zero-weight generator with only time-scale dependency (Eq. A1) followed a single-step calibration against observed zero-weight proportions by non-linear least squares.

- The parameters of the zero-weight generator with time-scale and depth dependency (Eq. A2) followed a 2-steps calibration: *i)* For each time scale, the proportion of zero-weight depending on depth was evaluated using a weighted running window to compensate for rare occurrence of extreme depths. The proportion of zero-weight depending on depth was then fitted to a function (Eq. A2a). *ii)* The functions modelling the parameters depending on time-scale were then calibrated by least square (parameters of Eq. A2b, A2c and A2d).

 – The parameters of zero-weight generator with time-scale, depth and temperature dependency (Eq. A3) followed a similar calibration procedure. *i)* Using running windows of temperature, the proportion of zero-weight depending on depth was fitted by least squares for different temperature (Eq. A3a). *ii)* Given a time-scale the parameters depending on temperature were fitted to a Gaussian function (Eq. A3b). *iii)* The parameters of the Gaussian function depending on time-scale were then fitted to set of functions by least square (Eq. A3c, A3d and A3e).

 – The non-zero-weight generator consisted a truncated normal distribution on [0,0.5] with $\mu = 0.5$ (Eq. 3b) and a function $\sigma$ depending on time-scale (Eq. A4). It was chosen against more commonly used beta distributions (McIntyre et al., 2016) after a goodness of fit test applied to the historical data. The calibration was done in 2 steps. *i)* $\sigma$ was evaluated by non-linear least squares for each time scale. *ii)* The parameters of Eq. A4 were calibrated against the evaluated $\sigma$ depending on time-scale by least square.

 – The parameters of the $SEP$ generator (Eq. A5) followed a single-step calibration by least square with processed proportion of high weight on the side of highest neighbour depending on time-scale.

## 2.3 Green Infrastructure modelling

In order to quantify the influence of rainfall input in green roof performance estimation, two green roofs located in Trondheim were modelled. They were selected due to data availability and the contrast of their behaviours: *i)* A typical extensive green roof (E-Green roof) with sedum vegetation, 30 mm of substrate, and 10 mm of "eggbox" drainage layer (Hamouz and Muthanna, 2019), and *ii)* a detention-based extensive green roof (D-Green roof) with sedum vegetation, 30 mm of substrate, and 100 mm of lightweight clay aggregates (Hamouz et al., 2020). The model (Eq. 5) was a simple reservoir model with smoothed linear function (Eq. 5c) for the outflow, Oudin's model for Potential Evapotranspiration ($PET$, Eq. 5b) and a Soil Moisture Evaluation Function to estimate Actual Evapotranspiration ($AET$) (Johannessen et al., 2017).

$$WC_i = WC_{i-1} + P_{i-1} - Q_{i-1} - WC_{i-1} \times PET_i \times C \tag{5a}$$

$$PET_i = \begin{cases} 0, & \text{if } T_i <= 5°\text{C} \\ \frac{Ra}{\lambda\rho} \times 0.01 \times (T_{mean} + 5), & \text{if } T_i > 5°\text{C} \end{cases} \tag{5b}$$

$$Q_i = \begin{cases} \frac{S_K}{1 + \exp(-\frac{4 \times K}{S_K} \times (WC_i - WC_K - \frac{S_K - 1}{2 \times K}))}, & \text{if } WC_i > WC_K + \frac{S_K - 1}{2 \times K} \\ K \times (WC_i - WC_K) + \frac{1}{2}, & \text{else} \end{cases} \tag{5c}$$

$$\tag{5d}$$

$WC_i$ is the water content (mm) at time $t_i$. $P_i$ is the precipitation (mm $\cdot$ min$^{-1}$). The discharge $Q_i$ (mm $\cdot$ min$^{-1}$) is based on the empirical (Eq. 5c). The temperature $T_{mean}$ is in Celsius degree, the extra-terrestrial radiation $Ra$ is derived from the latitude and the Julian day. The constant $\frac{1}{\lambda\rho} \approx 0.408$ depends on latent heat and volumetric mass of water. The factor $C$ is a calibrated factor depending on the maximum storage and the crop factor. The smoothed linear function (Eq. 5c) has

three parameters: $K$ the conductivity slope, $S_K$ the smoothing factor and $WC_K$ the starting delay. The model was developed based on data from extreme tests with artificial precipitation (Hamouz et al., 2020) by establishing a relationship between water content and runoff. The outflow depending on water content was used as input for calibration of the parameters of the discharge function using Bayesian calibration with DREAM setup (Laloy and Vrugt, 2012). It should be noted that the model remains limited as it lumps processes and neglects dynamical effect, i.e. the wetting of the aggregates and substrate and the spatial distribution of water content within the roof (Hamouz et al., 2020). The D-Green roof's model was tested against measured discharge with, as input, a rainfall series of two and half month from July 2018 to the $25^{th}$ of September, and a one-month series from the $5^{th}$ of September 2019 to the $5^{th}$ of October. The E-Green roof's model was tested against measured discharge with a rainfall series from April 2017 to September 2017 as input. Snow periods were mostly excluded for the evaluation.

## 2.4   Evaluation the downscaling models

For each location, the observed precipitations were aggregated to daily resolution and downscaled to obtain 200 time-series of 6-min time-step. They were used to model the extensive and detention-based extensive green roofs in parallel. It should be noted that irrigation needs and snow periods were neglected since the primary objective of the study was to evaluate the produced time-series. There were 10 projections available in Norway for the RCP 8.5 and 12 in France with the EURO-Cordex project. Each projected time-series was downscaled 20 times (200 simulations for Norwegian locations, and 240 simulations for French locations) to capture: *i)* the variability between the projections and *ii)* the variability due to the nature of the downscaling model. The number of simulations per location and per period was chosen to ensure reasonably low simulation time and represent the stochastic uncertainty inherent to the downscaling process. The stability of the percentile estimator with 200 simulations was verified against 1000 simulations in one model and one location to validate the choice.

To evaluate the performance of the downscaling model and the projected performance of green roofs, different indicators were used:

– The lag-1 autocorrelation depending on time-scale was evaluated. It was chosen to assess the temporal structure of the produced time-series. The autocorrelation depending on lag-time for time-scale 6-min, 48-min and 180-min where used for an in-depth analysis.

– The survival distribution of precipitation and discharge from both roofs were assessed at 6-min time-step. This approach is similar to the use of flow duration curves recently applied to green roofs by Johannessen et al. (2018). The exceedance probabilities were presented with a log axis to account for extreme probabilities. The median, $5^{th}$ and $95^{th}$ percentiles of the downscaled time-series were represented. The survival distribution of discharge from the roofs with downscaled time-series compared to the distribution based on observed data indicates the applicability of the downscaled time-series as an input for green infrastructure modelling.

– Along with the survival distribution, a performance indicator derived from the Kolmogorov-Smirnov (KS) distance was used. The KS distance was indeed not relevant for the survival distributions where the extreme probabilities are of prime importance. The authors did not find a standard indicator for such cases in the literature, therefore the following indicator,

that penalizes more errors for extreme probabilities, was developed:

$$KS_{rel} = max\left(\frac{Distrib_{Sim,median} - Distrib_{Obs}}{Distrib_{Obs}}\right) \tag{6}$$

– Three different discharge thresholds were used to report exceedance frequency on different operating modes: 1 L/s/ha for small events, 10 L/s/ha for major events and 100 L/s/ha for extreme events. Those thresholds were chosen in common for all roofs to facilitate comparison. They represent a compromise to have the same operating modes for each locations even if the occurrence of those modes differ due to different climate conditions. Small events duration were counted in days per year, major events in hours per year and extreme events in minutes per year.

– The distribution of dry periods and the retention fraction were computed. They are not expected to be affected by the downscaling process since the dry periods affecting the roofs can be observed on daily resolution, and the retention fraction can be estimated with conceptual models using daily time-step data. However, they provide additional information to analyse the behaviour of the roofs.

## 2.5 Hybrid event-based downscaling

In order to assess the applicability of downscaled time-series to predict the future performance of green infrastructure, the methods were compared to the current recommended practice in the locations: the use of an event-based design method based on $IDF$ curves with a climate factor ($CF$)(Kristvik et al., 2019). In particular, the variational method (Alfieri et al., 2008) is applied. It consists in, given a return period, to consider the constant-intensity rainfall leading to the highest discharge. It should be noted that the comparison intended to follow the recommended design method and not to follow the guidelines of a specific city since they can differ in terms of regulation. For instance, in Trondheim a threshold for maximum discharge has to be fulfilled (Trondheim Kommune, 2015) while in Lyon the 15 first mm of a 20-year return period has to be retained, and beyond those 15 mm a threshold is set for maximum discharge from the parcel (Greater Lyon council, 2020). The longest available time-series, originated from Trondheim, was the most adequate for this example. For 2, 5 and 10-year return period rainfall and runoff events, three approaches were compared: *i)* peaks runoff of runoff events based on an observed precipitation time-series (reference), *ii)* the peak runoff of rainfall events based on variational method, the IDF curves and with and without climate factor (typical design approach) and, *iii)* an hybrid approach based on downscaling $10^5$ rainfall events with a daily depth based on the return period curves with and without climate factors. This last approach used the $MRC_{SIT-SEP}$ model, the initial water content was set to the most probable value based on analysis of a long time-series. According to the current recommendation in Norway for Trondheim municipality, a climate factor of 1.4 was applied (Dyrrdal and Førland, 2019).

## 3 Result and discussion

### 3.1 Green infrastructure model

The parametrized empirical reservoir model was applied to the extensive green roof and the detention-based extensive green roof. The performance was evaluated both on the time-series and individual events extracted from the time-series. The criteria were: *i)* Nash-Sutcliffe Efficiency ($NSE$) indicator on time-series for both discharge and water content, *ii)* $NSE$ for rainfall events defined with a minimum inter events time of 6-hours to analyse further the behaviour of the model, and *iii)* the volumetric error on the time-series to account for model retention evaluation. The water content was estimated directly from discharge measurement using the empirical curve. The performance was as follows:

– $NSE > 0.8$ for both discharge and water content for the extensive green roof. On the 3 most intense events the $NSE$ ranged from 0.9 to 0.75. The water balance error was found to be 2.1%.

– $NSE > 0.94$ for both discharge and water content for the detention-based extensive roof. On the 3 most intense events the $NSE$ ranged from 0.96 to 0.85. The water balance error was found to be 5%.

The conceptual limitation of the model can be seen in Figure 2 at the beginning of the events of the testing period. It suggests that short events with low intensity are not reproduced well by the model as it cannot represent the delay induced by the wetting of the different layers of the roofs. Since the objectives of this study involve the use of a simple model to reproduce the behaviour of two roofs, the model was not further improved.

### 3.2 Analysis of climates properties

Figure 3 presents the zero-weight proportion depending on time-scale, depth and temperature for two different datasets (Bodø and Hamar). In Figure 3a the proportion of zero-weight decreases with increasing time-scale for Bodø. In Hamar the proportion decreases until 45 min and increases for higher time-scales. Based on this observation, two types of datasets were identified in terms of zero-weight occurrence. For data from Bodø, Bergen, Kristiansund and Trondheim, the proportion of weights that equalled zero decreased with increasing time-scale. For the data from Hamar, Lyon, Marseille and Kristiansand, the proportion decreased until 45 minutes time-scale and increased afterwards (Figure B1a). Given a time-scale, the proportion of weight equal to zero was not uniform depending on the weights (e.g. Bodø and Hamar Figure 3b with a time-scale of 48 minutes). Therefore, the monotony or non-monotony of the proportion of weights equalling to zero depending on time-scale can be explained by different distributions of depth in the observed data. The proportion depended on depth, which is consistent with previous work (Rupp et al., 2009). It should be noted that a high proportion of zero-weight is linked to shorter and more intense rainfall events. It could explain why the proportion is higher in Lyon than in Bergen (cf. appendix).

In Figure 3b, the zero-weights proportion decreases with increasing depth for the case of Bodø. In the case of Hamar, it increases for depth higher than 2 mm. Figure 3c and 3d show that a temperature dependency may explain this behaviour. In Bodø, the proportion depending on depth gives similar results for different ranges of temperature at 48-min resolution (Figure 3c). On the contrary, in Hamar, the subsets with lower temperature lead to a lower proportion of weights being equal

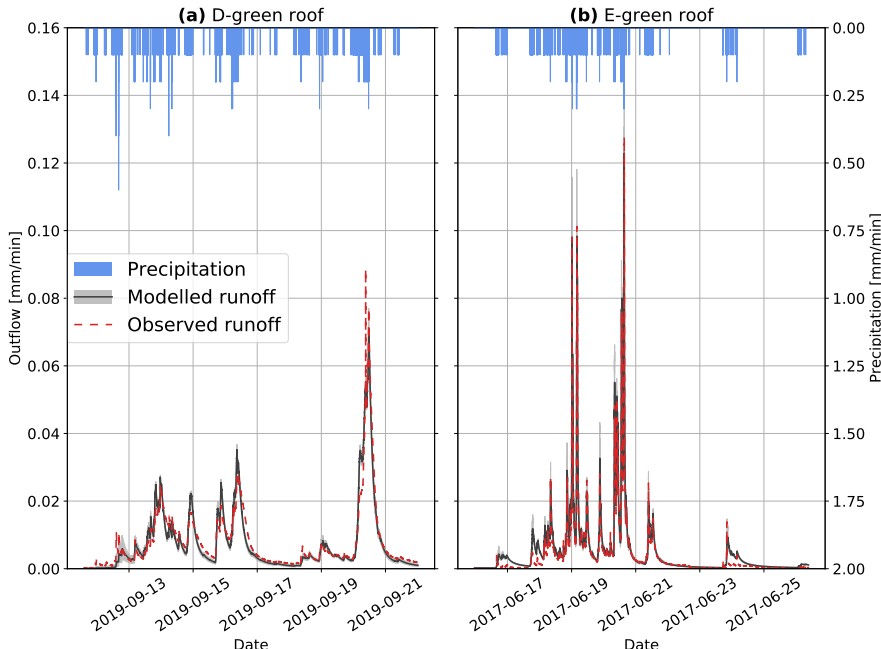

**Figure 2.** Testing of the green roof's reservoir model. Observed and modelled runoff of the detention-based extensive green roof (D) model on ten days period (left) and extensive green roof (E) for a period of eight days (right) in Trondheim.

to zero, compared to subsets with higher temperature Figure 3d). Moreover, the higher depths were observed in subsets with

higher temperature. The increase observed in Hamar can be explained by the distribution of observed values. It is consistent with the observation of different temporal distributions of rainfall for different temperature ranges such as convective rains (**?**Zhang et al., 2013). If, given a depth of 10 mm at resolution of 48 minutes, the probability to have a weight equal to zero is higher, then there is a higher probability to have an intense rainfall. The non-homogeneity of observed datasets and the shift in temperature with climate change might lead to inconsistency in time-series produced by the downscaling methods that exclude

depth and/or temperature dependency. The 48-min time-scale was chosen to exemplify this properties. The same properties can be observed for different time-scales but the magnitude differs and tend to lower with higher time-scale (Figure B1b, c and d). Developing a model without temperature dependency might prevent comparability of parameters between locations and does not necessarily lead to parameter parsimonious models. Moreover, a model such as $MRC_{SI}$ can result in overfitting when used with datasets like Hamar. The functions necessary to represent the behaviour without considering the temperature

dependency are more complex and less explanatory. Based on this analysis it was possible to add the temperature dependency and conceptualize a more explanatory model ($MRC_{SIT}$, with Eq. A3) with more robust results for the influence of climate change. This lies under the hypothesis that the information about the correlation between rainfall and precipitation will be expressed in the same way through those variables in the future.

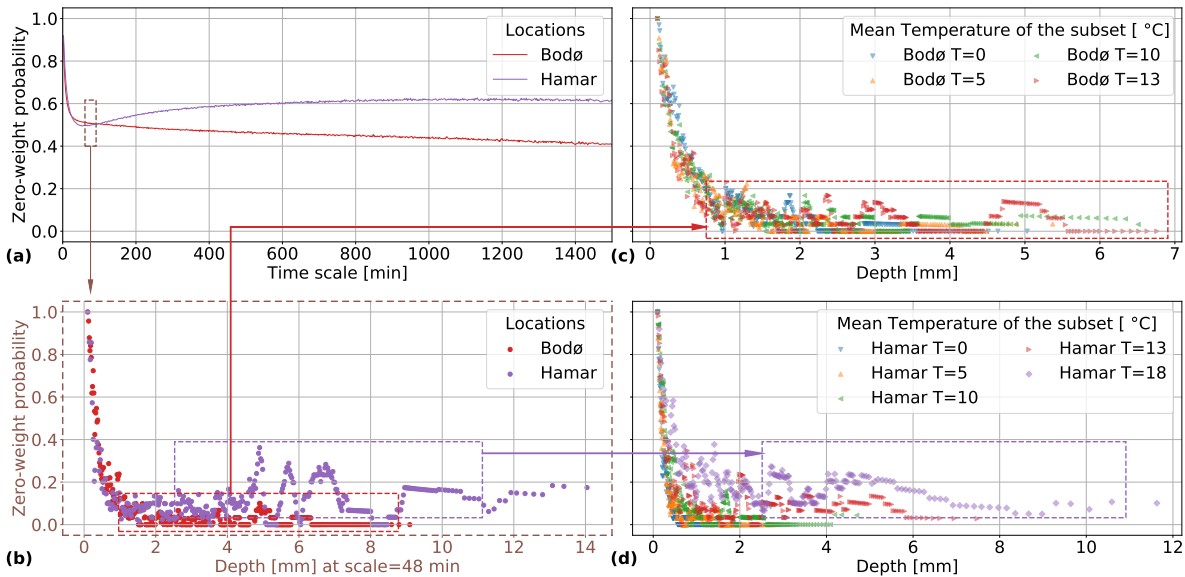

**Figure 3.** Dependency of the probability to have a weight equal to zero on: time-scale (a), rainfall depth (b) and temperature (c and d) for datasets observed in Bodø and Hamar. b, c, and d are based on data at 48-minute resolution.

### 3.3 Evaluation of the downscaling methods

An overview of the performance of the downscaling and green roof models in Bergen is presented on Figure 4. All the downscaling models performed similarly in terms of dry periods distribution and slightly underestimated the dry periods in observed data (Figure 4b). The dry periods were directly linked to the zero-weight probability. In green infrastructure modelling, the length of the dry periods influences the retention performance as it can lead to water stress hindering evapotranspiration. However, dry periods leading to water stress can be also evaluated with daily time-step series (there is no need for minute time-step series). Therefore, dry periods longer than the initial daily resolution are not significantly affected by downscaling.

The distribution of precipitation (Figure 4a) was properly reproduced by $MRC_{SI}$, $MRC_{SI-SEP}$, $MRC_{SIT}$, and $MRC_{SIT-SEP}$ ($KS_{rel} = 1$ in this case, indicating that the maximum distance has the same order of magnitude in data and model results) models while $MRC_S$ and $MRC_{S-SEP}$ underestimated low precipitation and overestimated high precipitation depths ($KS_{rel} = 10^2$ meaning that the maximum distance reached 2 orders of magnitude). This was expected as the time-steps with high depth have higher probability to not be split in the observed data. It is not the case for $MRC_S$ and $MRC_{S-SEP}$ models, which probability is uniformly distributed. In Bergen, the observed precipitations were within the range of 90% coverage interval for $MRC_{SI}$, $MRC_{SI-SEP}$, $MRC_{SIT}$, and $MRC_{SIT-SEP}$. For the four later mentioned models, the discharge of the D-Green roof was underestimated by one order of magnitude with a $KS_{rel}$ of $1.7 \cdot 10^1$ (Figure 4c), due to the behaviour of the roof with to rare high discharges. The hyetographs produced by downscaling probably tend to generate less favourable hyetographs for this roof. Although the discharge of the E-Green roof did not fall in the 90% coverage interval, it can be

considered as slightly underestimated since the magnitude is similar with a $KS_{rel}$ of 2.0 (Figure 4d). However, it was not the case for all locations, as in Hamar the most extreme precipitations tended to be underestimated while the discharge from both roofs had the same order of magnitude as the observed data but tended to be overestimated. These findings could suggest inconsistency in the temporal structure of rainfall. This hypothesis can be confirmed by the autocorrelation (Figure 5) being overestimated at 6 minutes time-step.

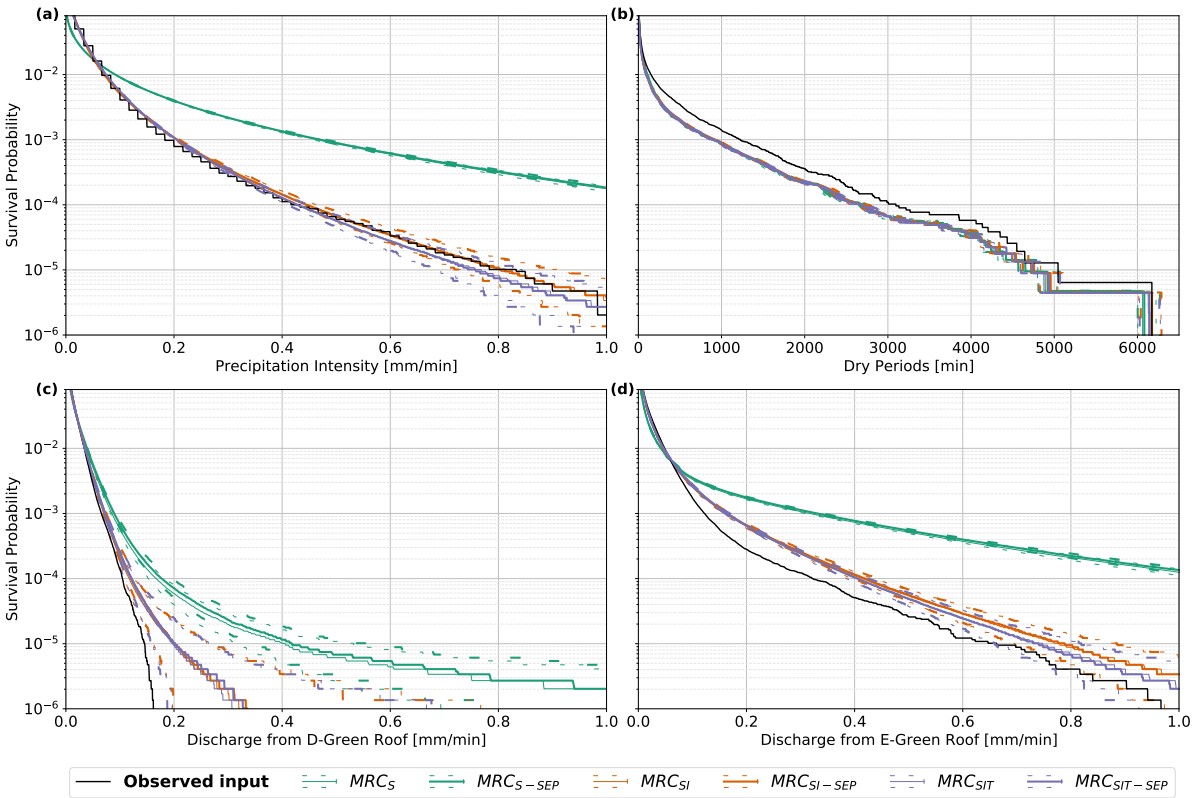

**Figure 4.** Models performance with data from Bergen current climate for $MRC_S$, $MRC_{S-SEP}$, $MRC_{SI}$, $MRC_{SI-SEP}$, $MRC_{SIT}$, and $MRC_{SIT-SEP}$ with a range from $5^{th}$ to $95^{th}$ percentiles. Observed input represents the fine-resolution observed time-series or simulation using this time-series as input.

The autocorrelation was underestimated by $MRC_S$ and $MRC_{S-SEP}$ models. The use of the rainfall continuity indicator increased the lag-1 autocorrelation for all models but did not improve the overall performances. The models $MRC_{SI}$, $MRC_{SI-SEP}$, $MRC_{SIT}$, and $MRC_{SIT-SEP}$ underestimated the lag-1 autocorrelation between 48 and 300 min time-scales, but an in-depth analysis with different lags at 48-min and 180-min time-scale shows that despite that underestimation for lag-1 the general behaviour of the observed time-series is reproduced. Similar observations were done for other locations.

To evaluate the produced time-series it is necessary to compare the discharge with observed time-series to the discharge with downscaled time-series. For most of the locations, the predicted range of precipitation or discharge deviated for lowest

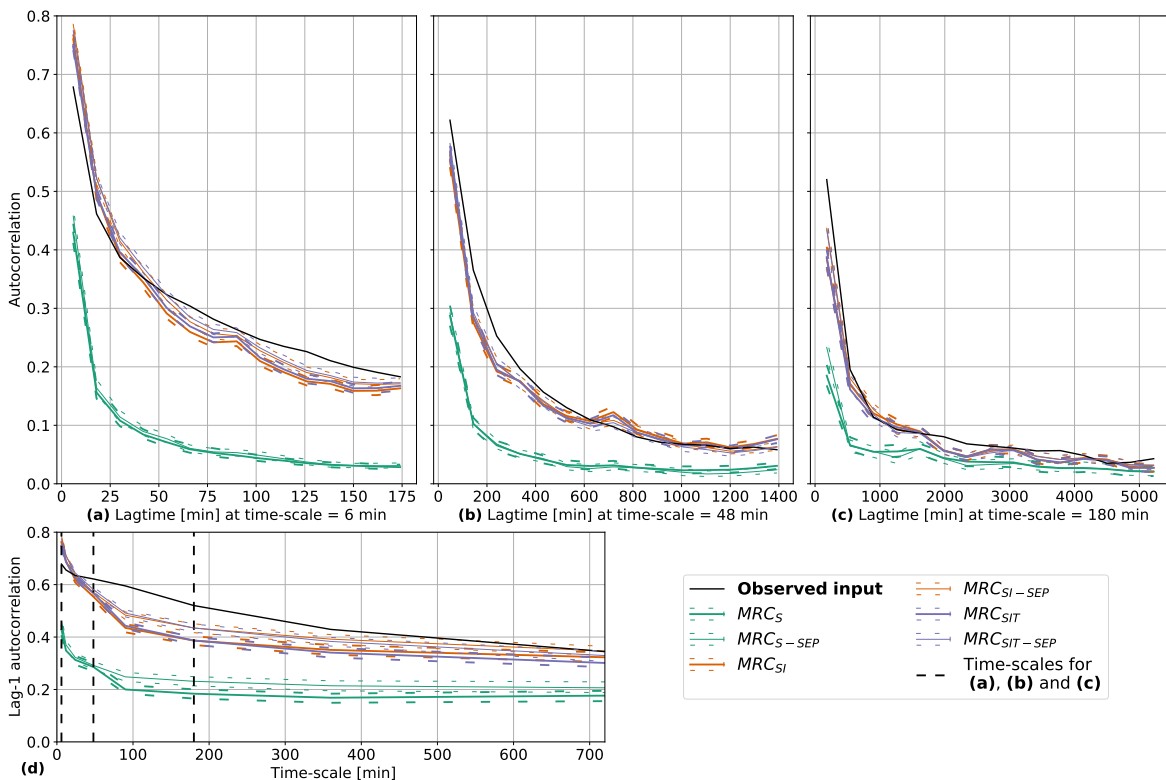

**Figure 5.** Autocorrelation with data from Bergen current climate for $MRC_S$, $MRC_{S-SEP}$, $MRC_{SI}$, $MRC_{SI-SEP}$, $MRC_{SIT}$, and $MRC_{SIT-SEP}$ with a range from $5^{th}$ to $95^{th}$ percentiles. Autocorrelation with different lags for 6-min, 48-min and 180 min time-scales, and lag-1 autocorrelation depending on time-scale. They are compared to the observed input which represents the fine-resolution observed time-series.

probabilities from the values obtained with observed time-series: *i)* when the precipitation range matched with the observed distribution, the discharge tended to be overestimated; *ii)* when the precipitation was underestimated, the discharge with ob-

served data tends to lay in the range obtained from downscaled time-series. While the downscaled time-series suffer from some limitation when compared to results obtained from the observed time-series, the raw discharge time-series might as well not be suitable for robust decision making in green infrastructure implementation as it does not represent the natural variation of performance of green infrastructure.

     In order to evaluate the potential of discharge from downscaled time-series to approach the range of performance linked

to natural variability, a 3-year moving window was used on precipitation and discharge time-series resulting from observed precipitation. The resulting $5^{th}$ and $95^{th}$ percentiles of the annual duration exceeding 1 L/s/ha, 10 L/s/ha and 100 L/s/ha are presented in Figure 6 to evaluate the time-series in different operating modes of the roofs. It is compared to the stochastic variability ($5^{th}$ and $95^{th}$ percentiles) from the 6 models. Each horizontal line in Figure 6 represents the range between $5^{th}$

and $95^{th}$ percentiles for the threshold and model considered. The different thresholds represent respectively discharge for small events, for major events and extreme events. In Figure 4, the threshold corresponds to 0.006 mm/min, 0.06 mm/min and 0.6 mm/min. A good estimate is defined by a complete or partial overlap between the observed natural variability and the stochastic variability range, the order of magnitude of the estimates should be similar. For instance, in Bergen (first column), the observed range of the E-Green roof higher than 100 L/s/ha (third row) is predicted, based on observed input, from 4 to 10 minutes; the $MRC_S$ model provided values around 200 minutes, it is not a good estimate as there is no overlap and the order of magnitude varies; the $MRC_{SI}$ model resulted in a range from 10 to 20 minutes. It is a good estimate as the ranges are overlapping, and the orders of magnitude are similar. The $MRC_S$ and $MRC_{S-SEP}$ models tend to underestimate the order of magnitude of the range of exceedance frequencies of the small events (1 L/s/ha) (in Bergen Hamar and Marseille) but tend to overestimate major (10 L/s/ha) (Hamar) and extreme events (100 L/s/ha) (Bergen Bodø Hamar and Marseille). The other models gave mostly good estimates for each of the thresholds (Figure 6, Figure C1). In Marseille, the models $MRC_{SI}$, $MRC_{SI-SEP}$, $MRC_{SIT}$, and $MRC_{SIT-SEP}$ tended to underestimate the higher bound of the extreme event precipitation with values lower than 50 minute per year whereas the observed time-series led to a maximum of 90 minutes per year. However, those models kept the order of magnitude, while $MRC_S$ and $MRC_{S-SEP}$ models estimated it higher than $10^2$ minutes. The same behaviour was observed with Hamar (Figure 6) and Lyon datasets (appendix, Figure C1). This suggests that the models performed worse for dryer locations, possibly due to the calibration procedure since less wet days are available for calibration. $MRC_{SI}$ and $MRC_{SIT}$ performed similarly, but due to its structure, $MRC_{SI}$ may overfit to the calibration data. It could result in an inaccurate prediction in case of significant temperature shift between the calibration and prediction datasets. To conclude, $MRC_S$ and $MRC_{S-SEP}$ lead to overestimation of the natural variability while $MRC_{SI}$, $MRC_{SI-SEP}$, $MRC_{SIT}$, and $MRC_{SIT-SEP}$ give more accurate estimates.

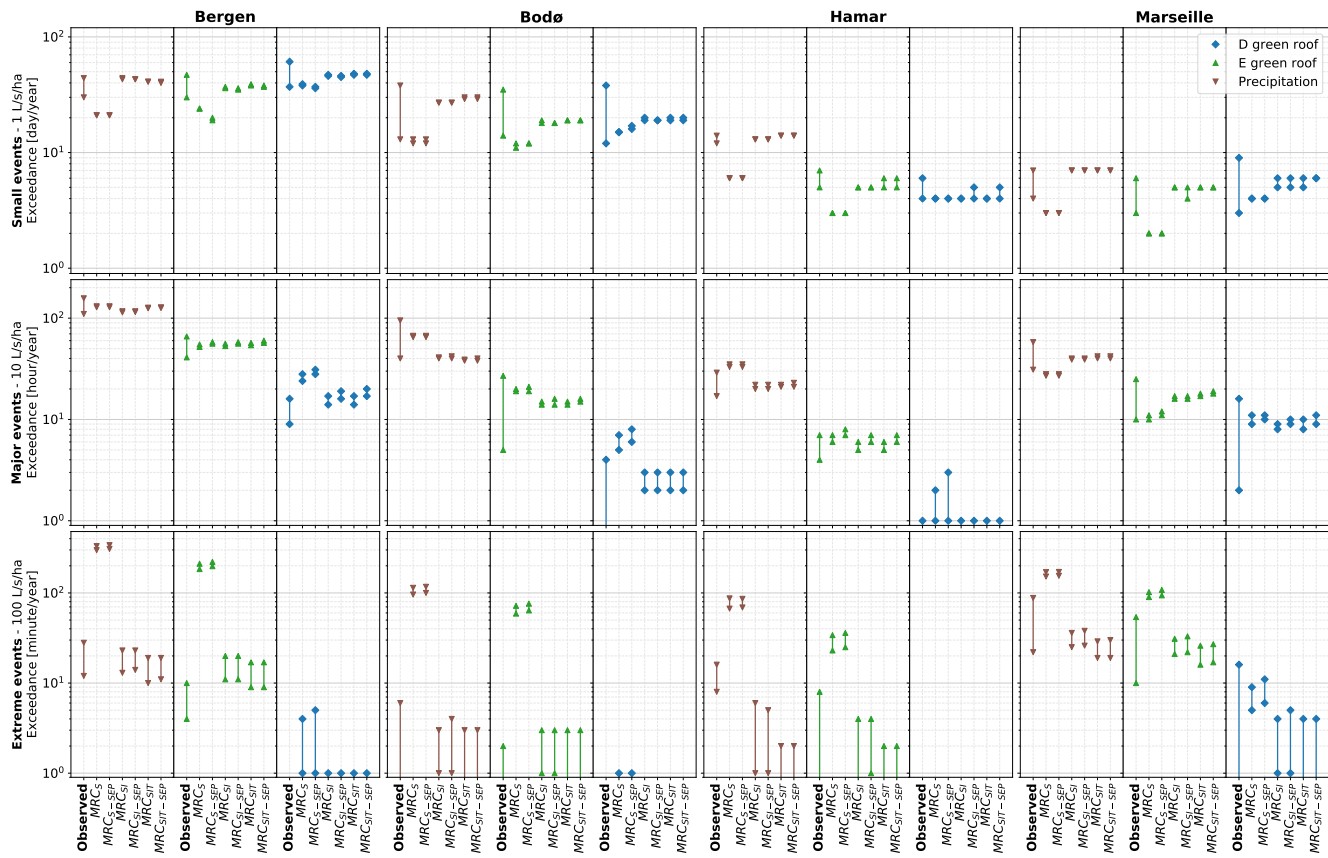

**Figure 6.** Performance of the downscaled time-series in Bergen, Bodø, Hamar and Marseille; exceedance frequency for small events, major events and extreme events. The stochastic variability linked to the downscaled time-series is evaluated with the $5^{th}$ to $95^{th}$ percentiles. *Observed* represents the fine-resolution observed time-series or simulation using this time-series as input; The $5^{th}$ to $95^{th}$ percentiles was estimated with a 3-year moving window. Due to log axis, occurences lower than $10^0$ are not visible.

### 3.4 Assessment of green roof future performance

All six models were used to assess the future performance of green roofs for future climate as illustrated for Bergen in
Figure 7. It was nevertheless acknowledged that $MRC_S$ and $MRC_{S-SEP}$ models gave less accurate estimates. The four
model $MRC_{SI}$, $MRC_{SI-SEP}$, $MRC_{SIT}$, and $MRC_{SIT-SEP}$ lead to similar results in Bergen (Figure 7). The difference
in estimates between the models with coherence indicator ($MRC_{SI-SEP}$, $MRC_{SIT-SEP}$) and without ($MRC_{SI-SEP}$,
$MRC_{SIT-SEP}$) was negligible in comparison to the stochastic uncertainty inherent to the models and the variability linked
to the different projections available under RCP 8.5 (Figure 7) . In Bergen, according to the projections, the performance of
the two solutions is likely to lead to worse performance: under the current climate, the 100 L/s/ha exceedance was lower than
1 minute for the D-Green roof; according to the $MRC_{SIT-SEP}$ model it might reach between 5 and 19 minutes in future
climate. It suggested a shift in the order or magnitude from $10^0$ to more than $10^1$ minutes. Similarly, the E-Green roof might
have a 100 L/s/ha exceedance shift from $10^1$ to $10^2$ minutes. It means that the threshold would regularly be reached.

As illustrated by Figure 8 and Figure C2, the performance shift depends highly on the location. While the 100 L/s/ha
exceedance of the green roofs was likely to get worse in Bergen, it was found to stay stable despite a small increase in Bodø
and to improve in Hamar and Marseille. The increase of exceedance frequency in the Norwegian cities was due to an increase
in precipitation (Table 2). However, the increase in temperature led to an increase in potential evapotranspiration and therefore
might have attenuated or even counterbalanced the effect of rainfall increase by lowering the initial water content in the roofs
at the beginning of a rainfall event. The Table 4 shows that the retention fraction was likely to decrease in Bergen, Bodø,
Hamar, Kristiansand and Kristiansund. It was found to increase in Lyon, Marseille and slightly in Trondheim. The models with
temperature dependency performed similarly to the model with only depth dependency in most of the location. However, in
Lyon and Marseille, the 100 L/s/ha exceedance or precipitation predicted differed from 16-27 min to 21-50 min (resp. 14-30
to 14-43 in Marseille). This suggests that some locations are more sensitive than other to temperature dependent patterns. The
models $MRC_{SI}$, $MRC_{SI-SEP}$, $MRC_{SIT}$, and $MRC_{SIT-SEP}$ allow to evaluate shift in performance for the different roofs
using exceedance range.

**Table 4.** Retention fraction in the different locations defined as the sum of outflow divided by the sum of precipitation.

| Location | Bergen | | Bodø | | Lyon | | Hamar | |
|---|---|---|---|---|---|---|---|---|
| Period | Observed | Projected | Observed | Projected | Observed | Projected | Observed | Projected |
| D-Green roof | 0.20 | 0.17 | 0.21 | 0.20 | 0.43 | 0.47 | 0.47 | 0.40 |
| E-Green roof | 0.19 | 0.16 | 0.21 | 0.20 | 0.39 | 0.44 | 0.44 | 0.38 |
| Location | Kristiansand | | Kristiansund | | Trondheim | | Marseille | |
| Period | Observed | Projected | Observed | Projected | Observed | Projected | Observed | Projected |
| D-Green roof | 0.24 | 0.22 | 0.25 | 0.20 | 0.27 | 0.30 | 0.41 | 0.47 |
| E-Green roof | 0.22 | 0.20 | 0.24 | 0.20 | 0.26 | 0.29 | 0.36 | 0.42 |

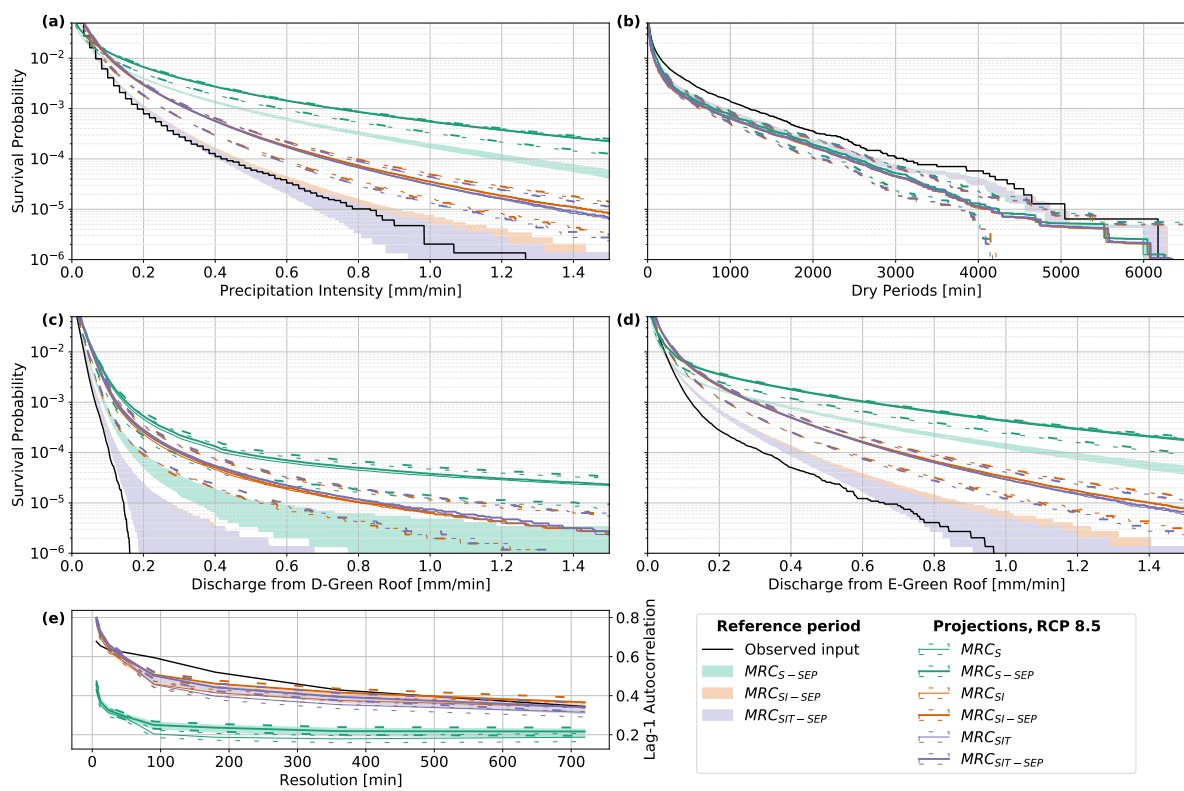

**Figure 7.** Comparison between performance under current climate and future climate in Bergen for the $MRC_S$, $MRC_{S-SEP}$, $MRC_{SI}$, $MRC_{SI-SEP}$, $MRC_{SIT}$, and $MRC_{SIT-SEP}$ with a range from the $5^{th}$ to $95^{th}$ percentiles. They are compared to Observed input which represents the fine-resolution observed time-series or simulation using this time-series as input

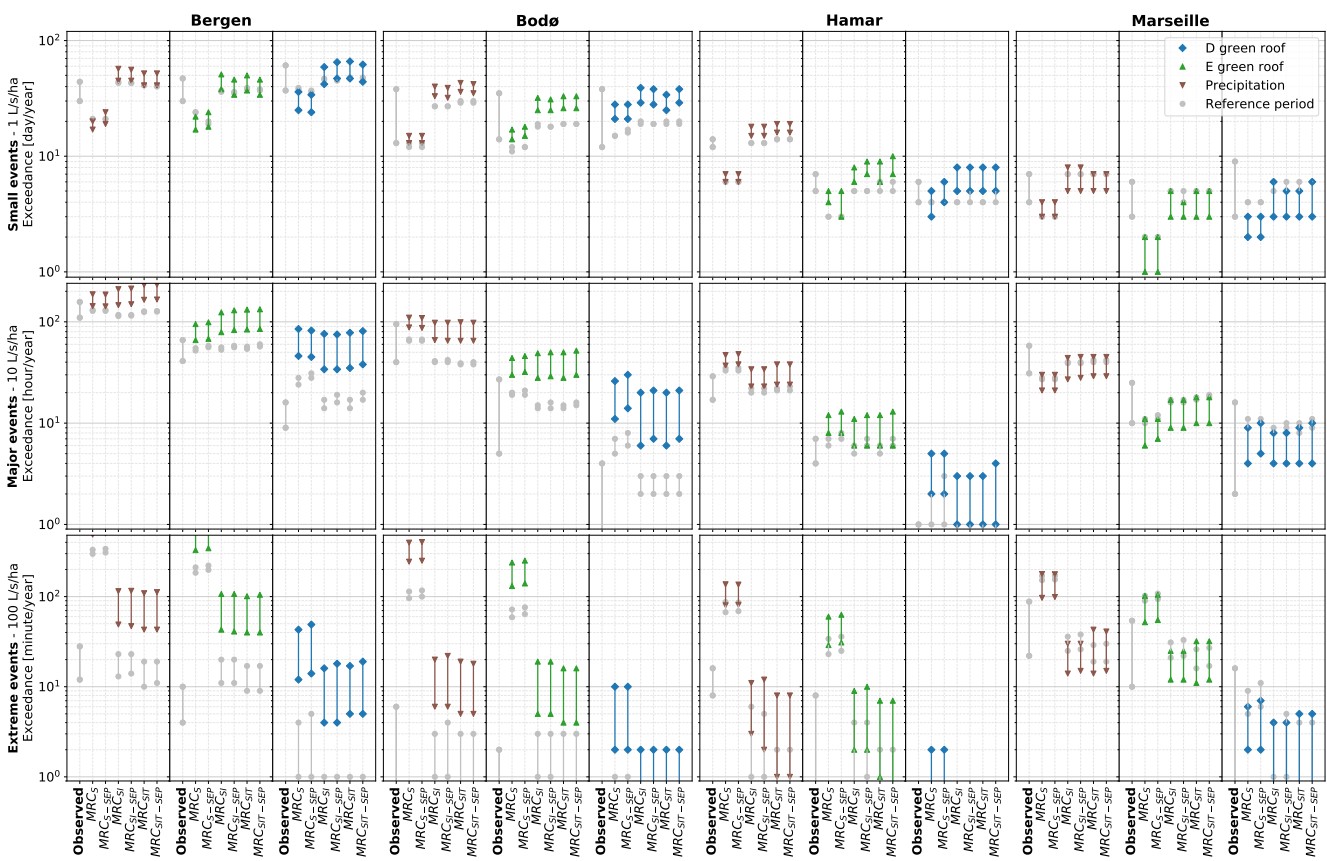

**Figure 8.** Future performance of green roofs (D and E) in Bergen, Bodø, Hamar and Marseille; exceedance frequency for small events, major events and extreme events. The stochastic variability linked to the downscaled time-series is evaluated with the $5^{th}$ to $95^{th}$ percentiles. $Observed$ represents the fine-resolution observed time-series or simulation using this time-series as input; The $5^{th}$ to $95^{th}$ percentiles was estimated with a 3-year moving window. Due to log axis, occurences lower than $10^0$ are not visible.

## 3.5 Design perspectives

The potential of downscaling models to improve the current practices was investigated. Figure 9 present results based on continuous simulation, on the variational method and on the hybrid approach with downscaled events. It shows that the variational method underestimated the peak runoff with observed data, and the distribution from the hybrid approach covered them. It suggests that the variational method might not be enough conservative when compared to peak runoff from runoff events instead of rainfall events. Even if the results from the hybrid event-based downscaling lead to realistic distribution based on probable rainfall events, the downscaling models might need a different calibration or conceptualization to be optimized specifically for extreme events. Moreover the initial water content for the events remain a limitation of this method. The observed peaks show a range of possible outcome which highlight the limitations of the variational method with a single estimate, whereas the hybrid downscaling-event based method, leading to a range of probable outcomes, gave promising results that can lead to more robust design and decision making. Due to its characteristics, the shift in performance between current climate and future climate is higher for the E-Green roof than for the D-Green roof. It is due to the detention layer in the D-Green roof which is not saturated by a 10-year return period event (Hamouz et al., 2020).

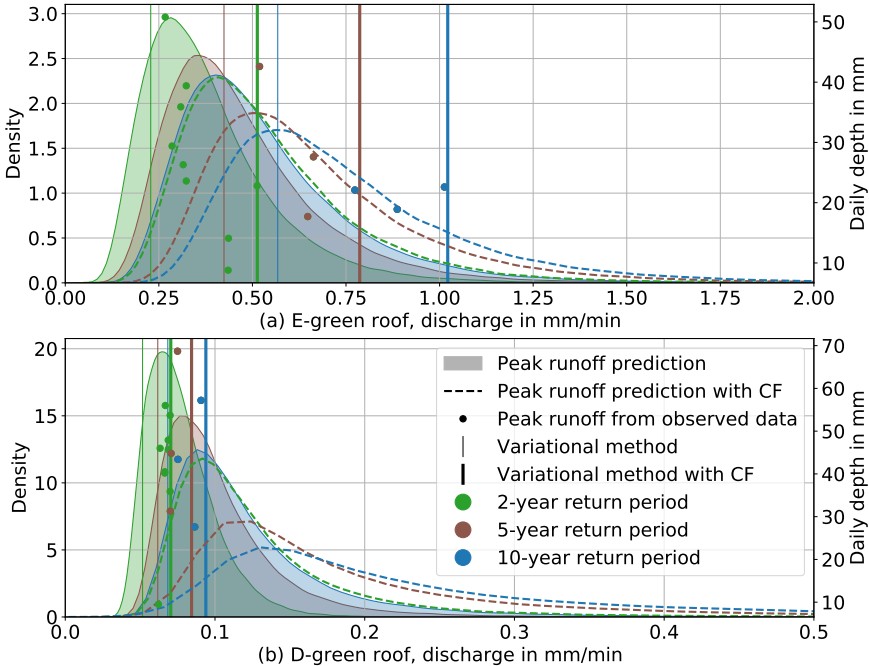

**Figure 9.** Performance depending on the return period in Trondheim for the extensive green roof (top) and the detention-based extensive green roof (bottom). The transparent coloured area (resp. dotted line) is the distribution based on the hybrid event-based downscaling under current climate (resp. with $CF$); the points represent the peaks runoff of runoff events from observed precipitation; the vertical lines the results found based on the $VM$. 2, 5 and 10-year return period are displayed.

# 4 Conclusions

In this study, multiplicative random cascades models with different variable dependency were developed. They were based on a study of time-scale, depth, and temperature characteristics of the datasets to ensure a consistent structure in the view to apply them to daily resolution climate projections. The applicability of the synthetic time-series to be used as input for performance modelling of green infrastructure was evaluated. They were used to predict the shift in runoff exceedance under a future climate.

Six downscaling model were developed: two models with only time-scale dependency ($MRC_S$ and $MRC_{S-SEP}$), two models with time-scale and depth dependency ($MRC_{SI}$ and $MRC_{SI-SEP}$) and two models with time-scale, depth and temperature dependency ($MRC_{SIT}$ and $MRC_{SIT-SEP}$). The models $MRC_{S-SEP}$, $MRC_{SI-SEP}$ and $MRC_{SIT-SEP}$ include a rainfall continuity property with the intention to improve the temporal structure of the rainfall. The parametrization of the models ensures the continuity of the different properties modelled and a low number of parameters.

The $MRC_S$ and $MRC_{S-SEP}$ were not sufficient to predict the future performance of green infrastructure as they lead to overestimation of runoff; The $MRC_{SI}$, $MRC_{SI-SEP}$, $MRC_{SIT}$, and $MRC_{SIT-SEP}$ lead to better performance: it was possible to predict runoff exceedance frequency with similar order of magnitude to an estimate of the natural variability of performance based on observed time-series. The structure of the $MRC_{SI}$ and $MRC_{SI-SEP}$ models make them more vulnerable to overfitting than $MRC_{SIT}$ and $MRC_{SIT-SEP}$ which make them less reliable for future performance estimate. However, the differences between them were negligible compared to the variability linked to the different outcome of climate models, the variability inherent to the model and its accuracy. The $MRC_{S-SEP}$, $MRC_{SI-SEP}$ and $MRC_{SIT-SEP}$ add an equation to improve the temporal structure of downscaled rainfall. The models predicted higher runoff from the detention-based extensive green roof, which is consistent with their properties, however the change in performance was not significant compared to stochastic uncertainty.

Using the RCP 8.5, the different downscaling and the green roof models suggests that the performance shift due to climate change highly depends on the location. The runoff exceedance is likely to increase in Bergen while slightly decrease in Lyon and Marseille and keeping the same order of magnitude in the other locations. The results were compared to one of the current practices: the use of the variational method with a climate factor. It highlighted the limitation of this practice that provide a singular estimate and underestimate the observed peaks. A hybrid method using downscaling on extreme events led to promising results by estimating a distribution of performance of peak runoff.

The models performed well in the 8 locations and 4 different climates. The use of a more advanced calibration procedure with Bayesian methods should improve the results. Similarly, a sensitivity analysis could improve the parametrization, especially for the models with depth and temperature dependency in order to fix non behavioural parameters. The current study does not include irrigation and snow modelling a study centred on green infrastructure modelling is therefore needed to extend the results. In order to be applied in practice on event-based simulation for design perspectives, the downscaling models needs to be improved with a calibration procedure developed for extreme events and not on the complete spectrum of observation as in the current study.

## Appendix A: Generators description

 ### A1   Zero-weight generator with only time-scale dependency

$$ZeroGen_S(S_{time}) = a_{14} \times \log(S_{time})^4 a_{13} \times \log(S_{time})^3 + a_{12} \times \log(S_{time})^2 + a_{11} \times \log(S_{time}) + a_{10} \tag{A1}$$

### A2   Zero-weight generator with both time-scale and depth dependency

$$ZeroGen_{SI}(d_{i,2j}, S_{time} = 2j) = \begin{cases} \frac{1}{1+d_{i,2j}-P_3(S_{time})} {}^{f_0(S_{time})} + \left(1 - \frac{1}{1+d_{i,2j}-P_3(S_{time})} {}^{f_1(S_{time})}\right), & \text{if } d_{i,2j} > a_2 \\ 1, & \text{else} \end{cases} \tag{A2a}$$

$$f_0(S_{time}) = \frac{S_{time}^{a_{00}-1} \times (1 + S_{time})^{-a_{00}-a_{01}}}{a_{02}} + a_{03} \tag{A2b}$$

 $$P_3(S_{time}) = b_{13} \times S_{time}^3 + b_{12} \times S_{time}^2 + b_{11} \times S_{time} + b_{10} \tag{A2c}$$

$$f_1(S_{time}) = \begin{cases} A \times (1 - \frac{1}{1+\exp(-\frac{4 \times B}{A} \times (WC_i - C - \frac{A}{2 \times B}))}, & \text{if } S_{time} > C - \frac{A}{2 \times B} \\ B \times (S_{time} - C), & \text{else} \end{cases} \tag{A2d}$$

### A3   Zero-weight generator with time-scale, depth and temperature dependency

$$ZeroGen_{SIT}(d_{i,2j}, T_{i,2j}, S_{time} = 2j) = \frac{1}{1+d_{i,2j}} {}^{gauss(T_{i,2j}, S_{time})} \tag{A3a}$$

$$gauss(T_{i,2j}, S_{time}) = A_0(S_{time}) \times \exp(\frac{(T_{i,2j} - \mu_T(S_{time}))^2}{2 \times \sigma_T(S_{time})^2}) \tag{A3b}$$

 $$\mu_T(S_{time}) = a_{14} \times S_{time}^4 a_{13} \times S_{time}^3 + a_{12} \times S_{time}^2 + a_{11} \times S_{time} + a_{10} \tag{A3c}$$

$$A_0(S_{time}) = \frac{b_0}{(1 + S_{time})^{b_1}} \tag{A3d}$$

$$\sigma_T(S_{time}) = \frac{c_0}{(c_2 + S_{time})^{c_1}} \tag{A3e}$$

### A4   Non-zero-weight generator

It consists in a truncated normal distribution described by Eq. 3b. The function $\sigma$ depends on time-scale:

 $$NonZeroGen_S(S_{time}) = a_{10} \times (S_{time})^{\frac{1}{a_{11}}} + a_{12} \tag{A4}$$

### A5   SEP generator

The Stochastic Element Permutation follow a function generating the threshold to be compared to a uniformly generated random number depending on time-scale:

$$SEPGen_S(S_{time}) = a_{14} \times \log(S_{time})^4 a_{13} \times \log(S_{time})^3 + a_{12} \times \log(S_{time})^2 + a_{11} \times \log(S_{time}) + a_{10} \tag{A5}$$

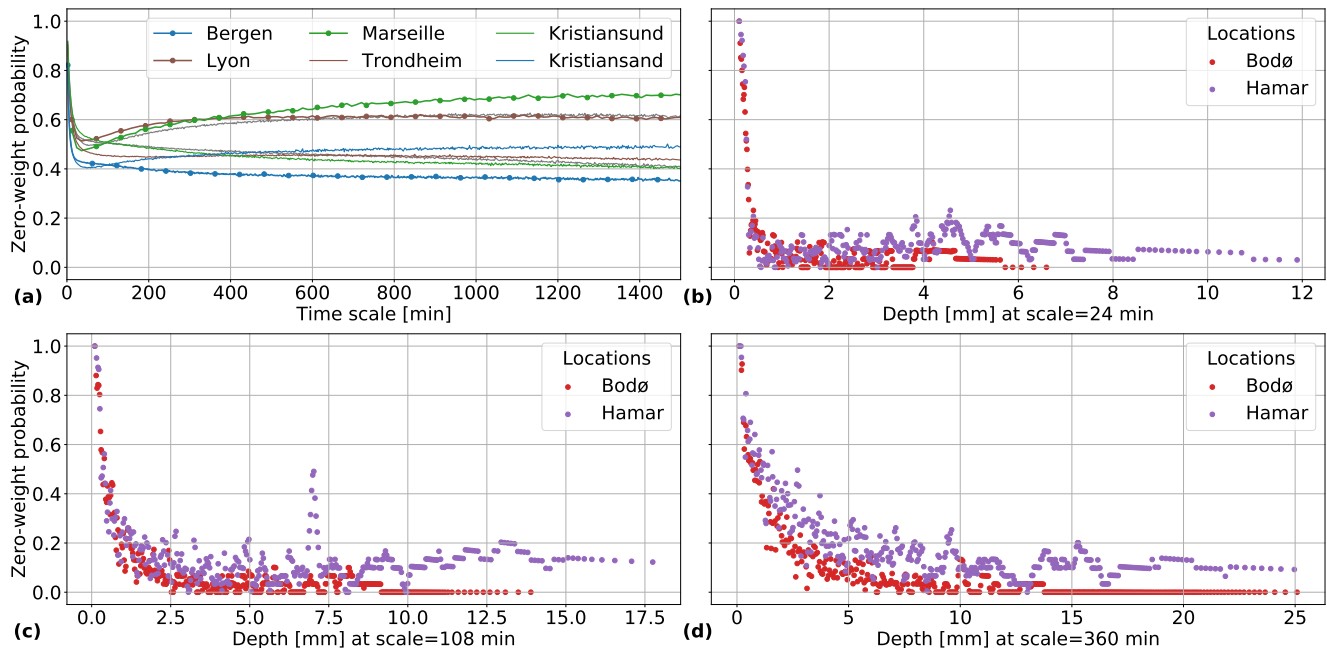

**Figure B1.** Zero-weight probability depending on time-scale for Bergen Lyon Marseille, Trondheim, Kristiansund and Kristiansand (a). Zero-weight probability depending on the rainfall depth for different time-scale: 24 min (b), 108 min (c) and 360 min (d) for Bodø and Hamar

# Appendix C: Other locations performance

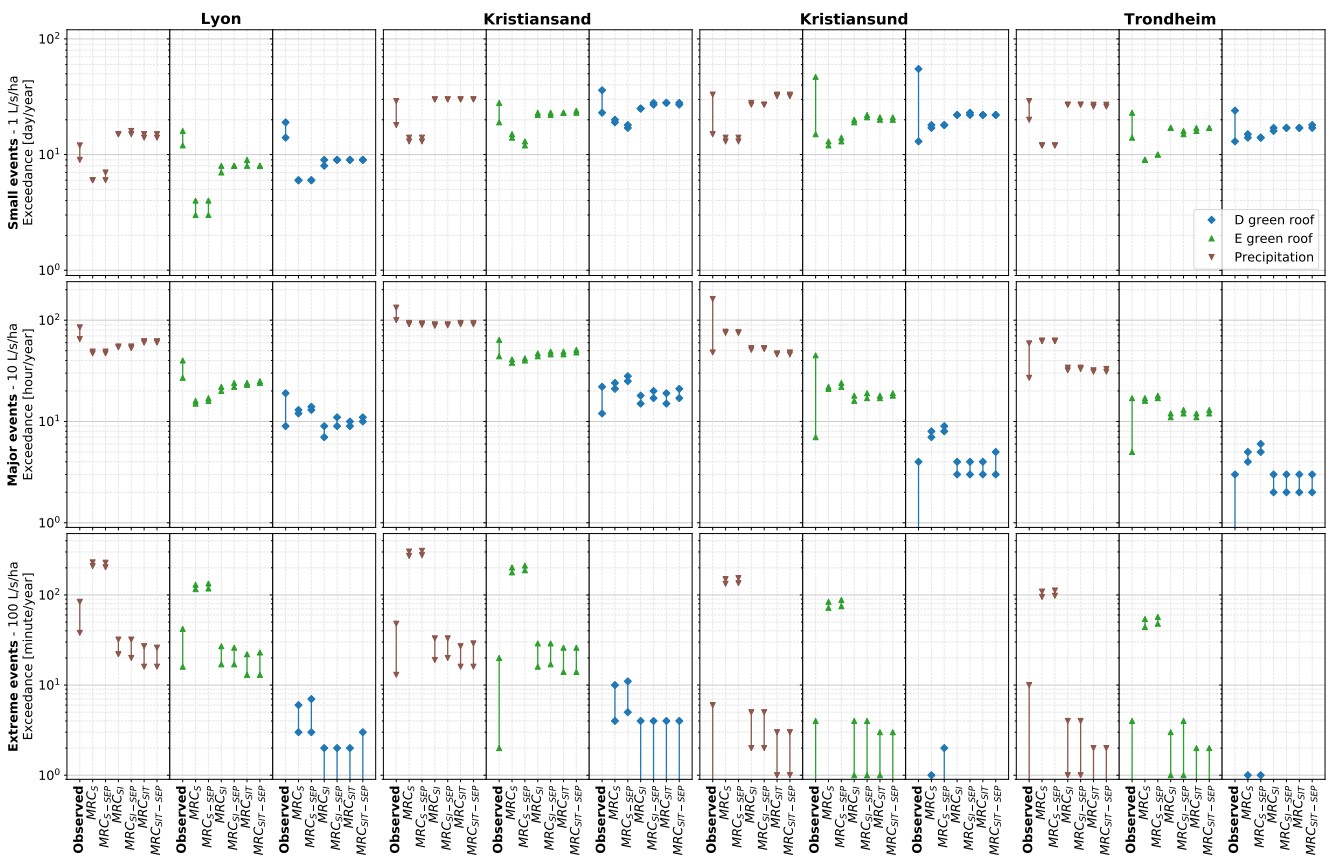

**Figure C1.** Performance of the downscaled time-series in Lyon, Kristiansand, Kristiansund and Trondheim; exceedance frequency for small events, major events and extreme events. The stochastic variability linked to the downscaled time-series is evaluated with the $5^{th}$ to $95^{th}$ percentiles. *Observed* represents the fine-resolution observed time-series or simulation using this time-series as input; The $5^{th}$ to $95^{th}$ percentiles was estimated with a 3-year moving window. Due to log axis, occurences lower than $10^0$ are not visible.

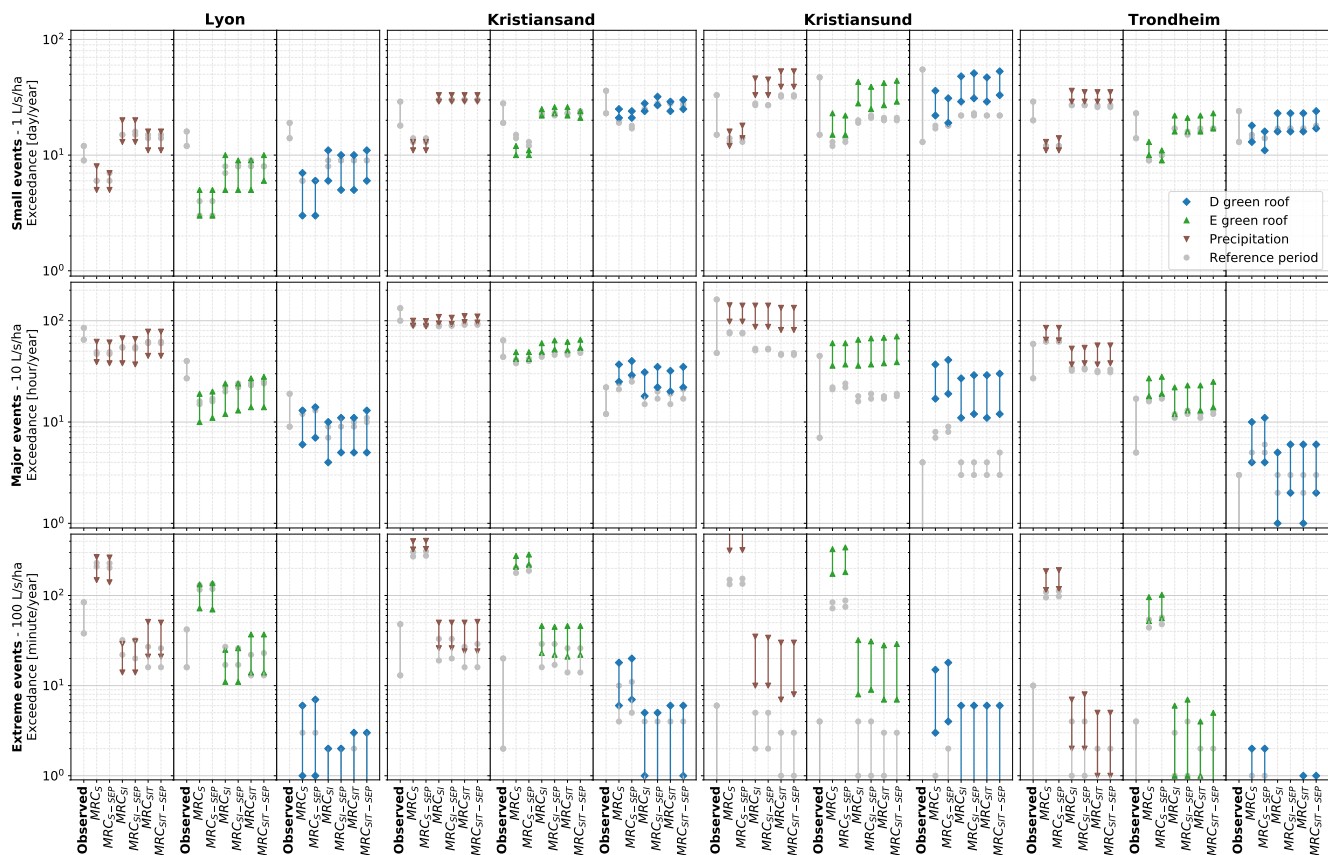

**Figure C2.** Future performance of green roofs (D and E) in Lyon, Kristiansand, Kristiansund and Trondheim; exceedance frequency for small events, major events and extreme events. The stochastic variability linked to the downscaled time-series is evaluated with the $5^{th}$ to $95^{th}$ percentiles. $Observed$ represents the fine-resolution observed time-series or simulation using this time-series as input; The $5^{th}$ to $95^{th}$ percentiles was estimated with a 3-year moving window. Due to log axis, occurences lower than $10^0$ are not visible.

*Author contributions.* Vincent was responsible for developing and programming the downscaling and green roofs models. Rasmus provided his expertise in downscaling. Jean-Luc came up with the idea of comparison with the variational method. The Norwegian meteorological institute, represented by Rasmus, provided the Norwegian data, Jean-Luc provided the French data. Tone, Edvard and Jean-Luc supervised each step of the study. Vincent wrote the first manuscript. The manuscript was revised by all co-authors.

*Competing interests.* The authors declare that they have no competing interests.

*Acknowledgements.* The study was supported by the Klima2050 Centre for Research-based Innovation (SFI) and financed by the Research Council of Norway and its consortium partners (grant number 237859/030). This research was partly performed within the framework of the OTHU (field Observatory for Urban Water Management - http://othu.org) and realized within the Graduate School H2O'Lyon (ANR-17-EURE-0018) and Université de Lyon (UdL) as part of the program "Investissement d'Avenir" ur by Agence Nationale de la Recherche (ANR). The authors acknowledge the Water Department of Lyon Metropole and the Water and Sewerage Department of Marseille Provence Metropole for sharing their rainfall data. The authors would like to thank the Norwegian Meteorological institute for providing the norwegian data and especially Cristian Lussana and Lars Grinde for their advices.

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
