# Peer review of "Forecasting green roof detention performance by temporal downscaling of precipitation time-series projections"

_Hydrology and Earth System Sciences, 2021_

## Editor Comment (EC1)

Review of Pons et al Temporal downscaling of precipitation time-series projections to forecast green roofs future detention performance.

Multiplicative random cascades is not within my field of research and I cannot therefore not perform a qualified review of this part. I will therefore leave it to the other two reviewers to detail herein. I have, however, worked quite a bit with climate projection of rainfall and downscaling issues related to urban hydrology where continuous rain series are required to simulate long term hydrological performance (see e.g. Thorndahl and Andersen (2021). Especially cases like the green roofs where the performance indeed depend on the antecedent conditions are interesting to investigate under future climate conditions.

The paper is generally well written but more details at the conceptual level could help the overall understanding of the proposed procedure.

I suggest describing more in detail the observational outflow data from the green roofs, the physical details of the green roofs, description of dominant processes of the roofs (e.g. in the introduction and not only in the method section), etc.

Table 3 shows the differences in retention fraction between observed and projected conditions. It could be interesting to see a similar table with observed values versus modelled performance for the current climate conditions in different MRC modes. Unlike fig 5 and 6, which are difficult to interpret,  a table summarizing the performances would clarify this part.

Is it possible to summarize the change of climate in some specific parameters, eg. changes in annual and seasonal precipitation, change in consecutive dry days, temperature, etc. for the different locations? This would help understand the differences between locations and maybe lead to an interpretation of the most important processes for the green roof performance and how the processes change under a change in eg. temperature, rainfall patterns, etc.

Best regards

Søren Thorndahl

Thorndahl, S., Andersen, C.B.(2021) CLIMACS: A method for stochastic generation of continuous climate projected point rainfall for urban drainage design; Journal of Hydrology, 602.

---

## Author Comment (AC1)

**R1_0**

*The manuscript "Temporal downscaling of precipitation time-series projections to forecast green roofs future detention performance" presented by Pons et al. describes the procedure to simulate rainfall through multiplicative random cascades. Six downscaling models, with time-scale, depth, and temperature dependency, have been developed to simulate time-series based on RCP8.5 to be applied on two green roofs with different properties and in different locations in France and Norway. Two green roofs have been modelled. Results show that the green roof performance shift due to climate change highly depends on the location.*

*Although the manuscript presents an innovative methodology and interesting results, it needs few minor adjustments and modifications before publication in an international journal such as HESS.*

> Thank you for this comment, we appreciate your interest in our manuscript. Modifications will be done according to the reviewers' comments in order to improve the quality of the current manuscript.

**R1_1**

*Green infrastructure, and in particular green roof, description could be more detailed in the introduction, explaining why green roofs have been chosen for this study and better highlighting the potential benefits of this work. Meaning and implications of RCP8.5 projections should also be included in the introduction.*

> We will add details about green infrastructure and green roofs. In particular, we will further highlight the necessity to model future performance of green infrastructure in consistence with the objectives of robust decision making and to achieve a resilient city from a stormwater management perspective.

> The green roofs have been chosen to be modelled for 3 main reasons: *i)* green roofs are one of the various Nature Based Solutions to be further developed in the next decades to contribute to a more integrated and sustainable urban water management, and thus exploring how their design and performance may be affected by future climate conditions is of particular interest. The work presented in this paper is part of a more global research project dedicated to the evolution and transition from existing grey infrastructures (sewer pipes) to blue-green infrastructures (NBS, etc.) or mixed infrastructures for future urban water management at the city scale, *ii)* both retention and detention are key processes in the hydrologic behaviour of green roofs, which implies that the adequate and realistic modelling of detention requires minute resolution time-series, generated by means of the downscaling approach, and *iii)* given the extreme *in situ* tests we performed (Hamouz et al., 2020), it was possible to calibrate a highly reliable and fast model accurate for both low and extreme events, which was of particular interest for simulation of future climate.

> About the potential benefits form this work in modelling future performance of green infrastructure, a paragraph about robust decision making and climate change adaption in the context of stormwater management will be added.

**R1_2**

*The use of abbreviations should be reviewed: sometimes abbreviations are not introduced, or not used. An abbreviation and symbol list could help the reader.*

> The abbreviation will be reviewed, and a table as you suggest will be added in order to help the reader.

| Abbrev. | Meaning | Change made | Reason |
|---|---|---|---|
| GI | Green infrastructure | - | - |
| MRC | Multiplicative Random Cascade | - | - |
| IDF curves | Intensity Duration Frequency curves | - | - |

| | | | |
|---|---|---|---|
| NVE | Norwegian Water Resources and Energy Directorate | - | - |
| MET | Norwegian Meteorological institute | - | - |
| S | Temporal coherence indicator at time-step $i$ and time-scale $2j$ | $S_{i,2j}$ | Avoid confusion |
| d | Depth at time-step $i$ and time-scale $2j$ | $d_{i,2j}$ | Avoid confusion |
| w | minimum weight at time-step step $i$ and time-scale $2j$ from aggregation of time-step $\{2i, 2i+1\}$ at time-scale $j$ | $w_{i,2j}$ | Avoid confusion |
| | | | |
| S | | | - |
| $MC$ | **MRC** model with only time**S**cale dependence | $MRC_S$ | - |
| $MCS$ | **MRC** model with time**S**cale dependence and **S**tochastic 2-**E**lement **P**ermutation | $MRC_{S-SEP}$ | - |
| $MCD$ | **MRC** model with time**S**cale and depth/**I**ntensity dependence | $MRC_{SI}$ | - |
| $MCDS$ | **MRC** model with time**S**cale, depth/**I**ntensity dependence and **S**tochastic 2-**E**lement **P**ermutation | $MRC_{SI-SEP}$ | - |
| $MCDT$ | **MRC** model with time**S**cale, depth/**I**ntensity and **T**emperature dependence | $MRC_{SIT}$ | - |
| $MCDTS$ | **MRC** model with time**S**cale, depth/**I**ntensity, **T**emperature dependence, and **S**tochastic 2-**E**lement **P**ermutation | $MRC_{SIT-SEP}$ | - |
| PET | Potential EvapoTranspiration | - | - |
| AET | Actual EvapoTranspiration | - | - |
| SMEF | Soil Moisture Evaluation Function | Removed | Used once |
| E-Green roof | Extensive green roof | - | - |
| D-Green roof | Detention based extensive green roof | - | - |
| WC_i | Water content in the roof at time i | Not in table | Equation variable |
| P_i | Precipitation depth at time i | Not in table | Equation variable |
| Q_i | Discharge released by the roof at time i | Not in table | Equation variable |
| T_mean | Mean daily temperature | Not in table | Equation variable |
| C | Calibrated factor accounting for Crop factor and maximum storage capacity | Not in table | Equation variable |
| S_K | Smoothing factor | Not in table | Equation variable |
| K | Conductivity slope | Not in table | Equation variable |
| WC_K | Starting delay | Not in table | Equation variable |
| DREAM | **D**iffe**R**ential **E**volution **A**daptative **M**etropolis | Not in table | Used once |
| RCP8.5 | Representative Concentration Pathway scenario with an 8.5 W/m2 radiative forcing in 2100 | - | - |
| NSE | Nash Sutcliffe Efficiency | - | - |
| VM | Variational Method | - | - |

*Figure 1: Review example of the different abbreviations*

**R1_3**

*The paper is overall well structured, although it could benefit from a review to catch typos (double or missing spaces etc.)*

We will review the manuscript in order to catch the typos.

**R1_4**

*[Abstract] Please avoid unnecessary abbreviations in the abstract*

Ok, unnecessary abbreviations will be removed from the abstract.

**R1_5**

*[1, 9-10] Please add an explanation for this result*

A synthetic explanation of this result will be added.

This is linked to local effect of climate change depending on cities. In general, in Norway the precipitation tends to increase, which leads to more frequent high values of initial conditions when a rain occurs. It thus tends to lead to a decrease of the capabilities of green infrastructure to manage stormwater under future conditions. In France the precipitation does not tend to increase while temperature increases. It leads to lower values of initial conditions in green infrastructure for day-to-day rain, which tends to increase their performance.

**R1_6**

*[1, 14] What's a?*

This typo will be fixed, sorry for the inconvenience (it should be 'a soil moisture evaluation function, and a crop factor'). Please also note that the two cited references on AET estimation equations are limited to the equations used in our model, other models can require more variables functions and data, which is beyond the scope of this paper.

**R1_7**

*[1 ,15] Evaporation processes time-scale could be higher than 24 hours, please add a reference to this sentence.*

This sentence refers to the data required to generate potential evapotranspiration estimates. Thornthwaite equation for instance can provide Daily or monthly estimate based on monthly temperature profile (Stovin et al., 2013). For the actual evapotranspiration, it is typical to use hourly (Stovin et al., 2013) or daily (Kristvik et al., 2019) precipitation values in the water balance solver. This information on daily data was mentioned here since future timeseries from euro-cordex are available at daily resolution which can be sufficient to estimate actual evapotranspiration. To avoid mis-interpretation, it will be clarified, and the sentence can be rewritten as follows: "In urban hydrology, typical time scales of processes are heterogeneous: from minutes to hours for rainfall, from hours to days for evapotranspiration".

**R1_8**

*[1, 20-24] What is the temporal resolution of these data? daily scale?*

Those data are available at daily scale, it will be clarified.

**R1_9**

*[2, 40] Please use the previously introduced abbreviation MRC*

The abbreviation will be used. As replied to comment R1_2, we will review the abbreviations and conduct a careful edit of the text to clarify their use.

**R1_10**

*[2 ,45] Reference written twice*

We will remove duplicate.

**R1_11**

*[2, 54] What is MC? This abbreviation has not been introduced before*

It refers to the 1st multiplicative random cascade model developed in this study which was introduced later. We will replace it by the general abbreviation MRC and clarify.

**R1_12**

*[2, Introduction] Please consider adding some background for green infrastructures and in particular for green roofs.*

It will be done, as detailed in reply to R1_1

**R1_13**
*[3, 66] and in Marseille? what data are available?*

In Marseille, similarly to Lyon data are available at 0.2 mm depth resolution with 6 -min time-step. It will be clarified. The rain gauge model, similar to the model in Lyon and most of the rain gauges used by Météo France, is a Précis-Mécanique tipping bucket rain gauge (1000 cm$^2$ and a tipping bucket of 0.2 mm capacity).

**R1_14**
*[3, 69] Climate projections in France have the same temporal resolution and length of the Norwegian projections? please specify*

The length of the timeseries depends on the climate model (GCM/RCM combination used to make available the timeseries from the EURO-CORDEX), some climate model only provides estimates until 2099. The lower boundary, 2071, was chosen at an earlier stage of the study to have a timeseries with similar length to the current timeseries and to focus on the last part of the century in order to apply it in a climate timeseries with a stronger shift in order to assess the robustness of the downscaling models. It was decided to use the same period for other locations. The time-scale is the same for all projections used in this paper: daily time-scale.

**R1_15**
*[5, 100] The acronyms used here have not been introduced before*

Indeed, they are introduced in the following sentence. Those are the names we decided to give to the different models. The table for acronyms will be provided and the paragraph will be clarified. A mentioned in R1_2, the name will be reviewed according to the table provided to clarify.

**R1_16**
*[8, 170-174] This paragraph fits better in the model description*

We agree that it could be debated were to place this paragraph. However, even though this model is a non-linear reservoir, its structure and calibration method (based on artificial rain) have not been introduced in previous paper. This is why the analysis of the model behaviour compared to the understanding of the green roofs from previous studies was presented in this section. Nonetheless the paragraph will be reviewed to clarify that the analysis is done by comparing the result of the validation period to the knowledge we have from previous field study; The sentence (l. 169) "The model is limited as it lumped processes and neglects dynamical effect: the wetting of the aggregates and substrate and the spatial distribution of water content within the roof (Hamouz et al., 2020)." does not depends on the results therefore can be moved.

**R1_17**
*[9, Figure 2] Please include in the figure the green roof type*

The green roof type will be added, and the figure will be improved.

**R1_18**

*[9, 175-185] Consider adding the results related to all locations in the appendix, to help the reader understanding the results*

We will add those results. More specifically, a new figure will be provided the appendix showing fig 3 a for all locations and fig 3 (b) for other locations and another scale (currently scale = 48 min). The authors agree that not only the location but also the scale might limit the understanding in the current manuscript.

**R1_19**

*[17, Paragraph 3.5] It would be interesting to have some considerations also for the French case studies*

The original objective of this part was to investigate the possible benefit of such a downscaling model from an application point of view. In the design perspective section, Trondheim was selected as one example among the 8 locations and linked to existing local guidelines. It is not our objective to proceed to the same application for the other cities, as in some of them, there are no detailed regulation or design rules for green roofs. For example, in Lyon, the stormwater regulation (dated 16 Dec. 2019) requires retaining the first 15 mm of rain and that, beyond these 15 mm, the maximum outflow from a parcel should be 3 L/s/ha, without indication of return period. In some specific areas in Greater Lyon with high risk of pluvial flooding, three levels are defined, corresponding to minimum 45, 55 and 70 mm of retention per event, or, respectively, lower retention if the infrastructure allows to manage the runoff for return periods of 5, 10 and 30 years. This Greater Lyon regulation remains unfortunately general, and no specific rule is directly suitable for the demonstration with a green roof that detain water. Indeed, the precipitation is delayed and attenuated but not permanently retained (except the evapotranspirated fraction which returns to the atmosphere).

Keeping the threshold defined in local guidelines for the city of Trondheim, the result using IDF curves for Lyon will be analysed and if it provides some relevant insight, it will be commented and added in appendix. In both case the choice of the location in this part will be clarified and discussed.

**Bibliography**

Hamouz, V., Pons, V., Sivertsen, E., Raspati, G. S., Bertrand-Krajewski, J.-L., & Muthanna, T. M. (2020). Detention-based green roofs for stormwater management under extreme precipitation due to climate change. *Blue-Green Systems*. https://doi.org/10.2166/bgs.2020.101

Kristvik, E., Johannessen, B. G., & Muthanna, T. M. (2019). Temporal downscaling of IDF curves applied to future performance of local stormwater measures. *Sustainability (Switzerland)*. https://doi.org/10.3390/su11051231

Stovin, V., Poë, S., & Berretta, C. (2013). A modelling study of long term green roof retention performance. *Journal of Environmental Management*. https://doi.org/10.1016/j.jenvman.2013.09.026

---

## Author Comment (AC2)

**R2_0**

*This paper investigates the use of downscaling model to forecast green roofs performance in the context of climate change. It uses a downscaling approach based on multiplicative cascades. The topic is interesting and relevant for the community. However, I would not recommend to publish this paper in its current state and suggest major revisions. Indeed, it requires significant clarifications on the downscaling model. Indeed, its presentation is hard to follow and should be more detailed.*

Thank you for your valuable comments. The original orientation of the paper was on developing downscaling model in order to apply them to GI models. That was the reason why the methodology was not fully detailed on the MRC model development. However, in the light of your comments, the authors agree that the methodology could be further detailed: i) by careful edits in the method section, and ii) by providing supplementary material such as model details. Please also note that one of the reasons why the python codes where not shared in the first version of the manuscript is because we plan to release it as a python package and making it available and stable does require more work than a direct code sharing.

**R2_1**

*It notably seems that different distributions of weights are used according to the cascade step suggesting they are not scale invariant.*

Indeed, the distribution is not scale invariant: the probability to get a weight equal to zero depends on time-scale (and possibly depth and temperature depending on the model). The non-zero weights follow a truncated normal distribution in which the sigma parameter depends on time-scale. In order to improve the manuscript, two main aspects will be developed: i) clarification in the method section, ii) details of the models (functions and structure) together with the python codes or a pseudo code corresponding to the models.

**R2_3**

*The calibration process of the numerous parameters (up to 19!) needs to be explained.*

The calibration procedure will be clarified. Please note that this procedure is a methodology for model development. For a wider use of the model, another methodology would be more appropriate: a more formal approach could be applied. It consists in different steps: see word file attached. The number of parameters, despite appearing high, is in fact still quite low compared to other micro-canonical cascades from 12 to 36 in total (Bürger et al., 2019) up to from 6 to 224 per disaggregation step (Müller-Thomy, 2020). The reason for that large number of parameters is that often a parameter set has to be estimated for each cascade steps. Similarly to Bürger et al., (2019), our models include timescale as a dependency, therefore there is a single (bigger) parameter set instead of a parameter set per cascade step.

The main idea of our calibration method is to first (**step 1**) calibrate for each time-scale, with moving window of depth and temperature. (**Step 2**) The timescale dependency is added by calibrating the parameter of step 1 depending on time-scale. The timescale dependency prevents for having a number of parameters at each cascade level which would lower the robustness of the model and reduce the number of parameters.

Taking the example of the MCDTS, without the time scale dependency, given 8 cascade step and 5 parameter per cascade step, there would be a total of 40 parameters. With the timescale dependency there is a total of 18 parameters. The model is them more flexible since it allows to use variable time-scale input data. The robustness of the model also improved using this procedure since for small time-scale the parameters are often noisy. See details below:

| | |
|---|---|
| A1 | Fit the proportion of zero-weight depending on time-scale to a function by non-linear least square. |
| A2 | Given a time-scale Fit the proportion of zero-weight depending on depth to a function by non-linear least square. Fit the parameters depending on time-scale to a function by non-linear least square. |
| A3 | Given a time-scale, given a window of temperature, fit the proportion of zero-weight depending on depth to a function by non-linear least square. Given a time-scale, fit the parameter depending on temperature to a gaussian function. Fit the parameters depending on time-scale to a function by non-linear least square. |
| B | Fit the distribution of non-zero weight to a truncated normal distribution on [0,0.5] with mu = 0.5 by fitting to the standard deviation of the sample. |
| C | Fit a function to the proportion of high weight on the side of the highest neighbour. |

| Model | Calibration steps |
|---|---|
| MC | A1, B |
| MCS | A1, B, C |
| MCD | A2, B |
| MCDS | A2, B, C |
| MCDT | A3, B |
| MCDTS | A3, B, C |

**R2_4**

*Please also clarify that what is called "observed data" for the various figures is actually simulations with observed rainfall. Am I correct?*

Exactly, it is simulation based on observed fine resolution time-series. It will be clarified.

**R2_5**

*- l. 40-44: It should clearly be stated that canonical cascades ensure conservation on average only while micro-canonical ones ensure exact conservation of intensity at each step.*

It will be clarified.

**R2_6**

*- l. 54: should MC be MRC? In general, the use of numerous abbreviations does not really help the reader. I would suggest limiting their use to words really often used in the paper.*

Yes, MC refer to the first model developed in this study. The abbreviation will be reviewed and a table for abbreviation will be provided according to the suggestions of the first reviewer.

| Abbrev. | Meaning | Change made | Reason |
|---|---|---|---|
| GI | Green infrastructure | - | - |
| MRC | Multiplicative Random Cascade | - | - |
| IDF curves | Intensity Duration Frequency curves | - | - |
| NVE | Norwegian Water Resources and Energy Directorate | - | - |
| MET | Norwegian Meteorological institute | - | - |
| S | Temporal coherence indicator at time-step $i$ and time-scale $2j$ | $S_{i,2j}$ | Avoid confusion |
| d | Depth at time-step $i$ and time-scale $2j$ | $d_{i,2j}$ | Avoid confusion |

| | | | |
|---|---|---|---|
| w | minimum weight at time-step step $i$ and time-scale $2j$ from aggregation of time-step $\{2i, 2i+1\}$ at time-scale $j$ | $w_{i,2j}$ | Avoid confusion |
| S | | | - |
| $MC$ | **MRC** model with only time**S**cale dependence | $MRC_S$ | - |
| $MCS$ | **MRC** model with time**S**cale dependence and **S**tochastic 2-**E**lement **P**ermutation | $MRC_{S-SEP}$ | - |
| $MCD$ | **MRC** model with time**S**cale and depth/**I**ntensity dependence | $MRC_{SI}$ | - |
| $MCDS$ | **MRC** model with time**S**cale, depth/**I**ntensity dependence and **S**tochastic 2-**E**lement **P**ermutation | $MRC_{SI-SEP}$ | - |
| $MCDT$ | **MRC** model with time**S**cale, depth/**I**ntensity and **T**emperature dependence | $MRC_{SIT}$ | - |
| $MCDTS$ | **MRC** model with time**S**cale, depth/**I**ntensity, **T**emperature dependence, and **S**tochastic 2-**E**lement **P**ermutation | $MRC_{SIT-SEP}$ | - |
| PET | Potential EvapoTranspiration | - | - |
| AET | Actual EvapoTranspiration | - | - |
| SMEF | Soil Moisture Evaluation Function | Removed | Used once |
| E-Green roof | Extensive green roof | - | - |
| D-Green roof | Detention based extensive green roof | - | - |
| WC_i | Water content in the roof at time i | Not in table | Equation variable |
| P_i | Precipitation depth at time i | Not in table | Equation variable |
| Q_i | Discharge released by the roof at time i | Not in table | Equation variable |
| T_mean | Mean daily temperature | Not in table | Equation variable |
| C | Calibrated factor accounting for Crop factor and maximum storage capacity | Not in table | Equation variable |
| S_K | Smoothing factor | Not in table | Equation variable |
| K | Conductivity slope | Not in table | Equation variable |
| WC_K | Starting delay | Not in table | Equation variable |
| DREAM | DiffeRential Evolution Adaptative Metropolis | Not in table | Used once |
| RCP8.5 | Representative Concentration Pathway scenario with an 8.5 W/m2 radiative forcing in 2100 | - | - |
| NSE | Nash Sutcliffe Efficiency | - | - |
| VM | Variational Method | - | - |

*Figure 1: Review example of the different abbreviations*

**R2_7**

- Section 2.2.1: I think there is a need to be more specific, notably for the reader not specialist. Index i and j should be consistent between equations 1 – 2 and Fig. 1. Please also clarify the range of possible values (if "i" refers to a time step then it belongs to 1… 2^n where n is the cascade step and j * 2^n = total duration?). Eq. 2: S is said to measure a proportion while it has only 3 possible values. Please clarify.

> We will clarify and be more consistent in the naming. "i" will refer to a time step and j a time-scale in minute. The first step of the cascade n=0 allow to go from2j=1440 minute to j = 720 minutes. You can see below a formal version of figure 1. The improved version will include those aspect together to a better readability for non expert reader.

[Figure]

Figure 2: Formal version of figure 1 in the first version of the manuscript.

**R2_8**

*- Section 2.2.2: Please clarify how the fitting of the models was done. Is the probability distribution used the same at all cascade steps (only for P(W=0) if I understand well table 2)? Was a scaling break identified in the data? What would be the consequences of such break? l. 101: "all included 5", may be say all included the use of eq. 5 to help the reader. Please explain how the depth or temperature dependency was included. It would also be needed to clarify in Table 2 to what refer the parameters mentioned.*

About the fitting description, please see reply to R2_3. The distributions and functions depend on the time-scale, it is therefore different at each cascade step. In practice the parameters for the distribution vary more at small time-scale than (approx. lower than 45 min time-scale) as it can be seen in the manuscript on figure 3.a. The consequence of such a break is that it is relevant to add time-scale dependency since *a priori*, given 2 different time-scales the distribution is not the same.

It will be clarified both in the text and together with supplementary material and details on the model (R2_1). Table 2 was corrected due to wrong version leading to confusion (see below).

| Model | $P(W=0)$ | | | | $CDF(W,W\neq 0)$ | | | | $P(S_W=high)$ | | | | Number of parameters |
|---|---|---|---|---|---|---|---|---|---|---|---|---|---|
| | R | D | T | N | R | D | T | N | R | D | T | N | |
| 1-MC | x | | x | | | | | | | | | | 8 |
| 2-MCS | x | | x | | | x | | | | x | | | 13 |
| 3-MCD | x | x | x | | | | | | | | | | 14 |
| 4-MCDS | x | x | x | | | x | | | | x | | | 19 |
| 5-MCDT | x | x | x | x | | | | | | | | | 13 |
| 6-MCDTS | x | x | x | x | | x | | | | x | | | 18 |

| Model | $P(W=0)$ | | | | $CDF(W,W\neq 0)$ | | | | $P(S_W=high)$ | | | | Number of parameters |
|---|---|---|---|---|---|---|---|---|---|---|---|---|---|
| | R | D | T | N | R | D | T | N | R | D | T | N | |
| $MRC_S$ | x | | | | x | | | | | | | | 8 |
| $MRC_{S-SEP}$ | x | | | | x | | | | x | | | x | 13 |
| $MRC_{SI}$ | x | x | | | x | | | | | | | | 14 |
| $MRC_{SI-SEP}$ | x | x | | | x | | | | x | | | x | 19 |
| $MRC_{SIT}$ | x | x | x | | x | | | | | | | | 13 |
| $MRC_{SIT-SEP}$ | x | x | x | | x | | | | x | | | x | 18 |

*Figure 3: Current table with wrong column alignment (top) table with alignment correction (bottom)*

The table 2 will be reviewed to clarify that all process involves a time-scale dependency.

The detail of the different models (including functions) will be provided in appendix.

**R2_9**

*- l. 144-145: why limiting to lag-1?*

In the manuscript the authors made the choice of applying a lag-1 autocorrelation at each step of the cascade. The main reason is that, at fine resolution, the autocorrelation might be influenced by the rain gauge resolution. The computational time is also shorter. Informally, the lag-1 at 90 min includes information relative to lag 2 or lag 3 at 45 min resolution, the same principle can be applied for all timesteps.

Since the autocorrelation is often computed for other lags (Müller-Thomy, 2020), we will further investigate other lag times and if relevant include it in the main text, or in an appendix.

**R2_10**

*- l. 152-153: how do you define "small", "major" and "extreme" events?*

We defined "small", "major" and "extreme" with common threshold for roofs and locations. In order to qualify different operating mode of the roofs and different climates while being common for all locations in order to facilitate the comparison, we had to set a compromise: 1 L/s/ha, 10 L/s/ha and 100 L/s/ha. Since one of the indicators is exceedance frequency, a common frequency could not be chosen as threshold.

We will clarify the choice in the method section.

**R2_11**

*- Section 3.2: How to you interpret physically the differences of behaviour in Fig. 3.a? Is the shape of Fig. 3.b the same for other time scales? How was the fitting of the model done from this analysis?*

In this context the shape in figure 3.a. cannot be linked to a physical behaviour but to the properties of the datasets. Those properties in the datasets can be linked both to a data collection problematic or to a physical property. In general, this curve is linked to the probability to have a long continuous evenly distributed event or a shorter event. The shape for figure 3b is the same for other timesteps, t=48 was chosen as an example since the effect is visual a support the explanation. The conceptualization of the Temperature dependent models was based on this analysis. cf. R2_3 for fitting. A figure will be provided in appendix (cf. R1_18: fig 3a for all locations and fig 3 (b) for other locations and another time-scale).

**R2_12**

*- l. 212-213 and comments on Fig. 4. c and d. Why is the discharge considered to be only slightly underestimated while the observations do not fall in the 5-95% percentile?*

The discharge looks are qualified as slightly underestimated because the distance between the distributions is small. The log axis was used here to track visually the magnitude to this underestimation. The reason for the use of log axis is that the exceedance frequency we are interested in are rarely occurring. That is also the reason why a Kolmogorov Smirnov test is not relevant here: we are interested in reliable metrics for rare occurrences.

In practice, those survival distributions are used to estimate the time above threshold. And from a practitioner point of view, the authors think that accounting for natural variability with a window of time is more relevant than looking at a single point estimate from a full time series since the duration associated to discharge exceedance frequency can vary from year to year, especially because of the rare occurrence of extreme events.

The figure 5 shows in 3 different thresholds that, while accounting for natural variability on the flow duration curve, the results from this method are close enough to inform on the magnitude of runoff occurrence. We will provide statistical distance (and possibly provide a Q-Q plot in appendix) to support this analysis. Showing the range of this 3 year-window from both observed and simulated data was initially excluded to favour readability of figure 4.

We will rework the paragraph to introduce this definition of good estimate earlier in order to clarify those aspects. We will also consider a way to add this information into the graphs 4.

**R2_13**

*- l. 216-218 and fig 4.e: the discrepancies between models and observations for the lag-1 autocorrelation should be discussed more.*

The MC led to poor lag-1 autocorrelation depending on time-scale because the depth is not taken into account when splitting the data. therefore, a high depth will be split in 2 in the same way as a small depth. It directly influences the lag-1 autocorrelation. About the MCDS and MCDTS, it is possible to improve the lag-1 autocorrelation. However, since the Rainfall continuity indicator does not take into account the depth of neighbouring approach, it was not possible to improve further the quality of the autocorrelation. See (Müller-Thomy, 2020) for other possibilities in improving autocorrelation (excluded to not increase further the number of parameter)

**R2_14**

*- l. 235-238: I have trouble to find the figures mentioned in Fig. 5.*

The results are indeed not easy to read on this because of the large amount of data and variable. The current example refers to Bergen **[1st column]**, major events (i.e., 10 L/s/ha) **[2nd**

**row]**, and D-green roof **[right part of the subplot]**. Moreover, this example is not the easier to read. Therefore, we will: *i)* carefully review the explanation linked to figure 5, 7 A1 and A2; *ii)* we will choose a clearer example. E.g., Bergen [1st column], small event [1st row] E-green roof [middle of the subplot] predicted between 30 to 45 days per year above threshold with simulation based on observed data. With MCS we predicted 18 to 20 day / year: it is a bad estimate (according to definition l. 235). With the MCDS we predicted 35 to 37 day/year: it is a good estimate. We will also consider changing the scale to a log scale in order to make the results more easily readable (see comment below).

**R2_15**

*- Fig. 5: last row (extreme events). May be the vertical scale could be split to enable a zoom on the lower part which concentrate most of the information which is not visible now.*

This figure was originally made in order to include the results from one location for each of the climate investigated and each of the events defined which made it challenging to include all. We will try both your suggestion and the use of a log axis (since the magnitude of the estimates matters especially).

**R2_16**

*- Fig. 6: "observed data" is not visible in the graphs*

It will be fixed.

**R2_17**

*- Section 3.5: without explaining everything, I believe that some details the variational approach are needed for the non-specialist reader. How do authors interpret the fact that the differences between the two approaches are much more pronounced extensive roofs than the detention-based ones?*

More details will be provided. Section 2.4 will be renamed "Evaluation of the downscaling models", and the use of the variational method will be added. It consists, given an IDF curve, in using as an estimate the constant duration rainfall leading to the worst-case scenarios. In our case, in terms of peak discharge.

The authors are not sure what the reviewer means by the 2 approaches: current climate vs future climate or downscaling based vs VM? For both cases, the difference between the approaches can be explained by the properties of the roofs (which is also the reason why they were selected). The D-green roof has a higher detention capacity which results in a very narrow distribution. However, as it can be seen with 10-year RP future, once this capacity is overcome (i.e., the layer saturated), it leads to much higher runoff and therefore a more spread distribution of performance. The E-green roof has a simple setup, it led more easily to high runoff and therefore it has a large range of performance even for lower return period events. However, increasing the return period shifts the range but does not increase the range as much as the E-green roof.

**References:**

Bürger, G., Pfister, A., & Bronstert, A. (2019). Temperature-driven rise in extreme sub-hourly rainfall. *Journal of Climate*. https://doi.org/10.1175/JCLI-D-19-0136.1

Müller-Thomy, H. (2020). Temporal rainfall disaggregation using a micro-canonical cascade model: possibilities to improve the autocorrelation. *Hydrology and Earth System Sciences*, *24*(1), 169–188.

---

## Author Comment (AC3)

**R3_0**

*Multiplicative random cascades is not within my field of research and I cannot therefore not perform a qualified review of this part. I will therefore leave it to the other two reviewers to detail herein. I have, however, worked quite a bit with climate projection of rainfall and downscaling issues related to urban hydrology where continuous rain series are required to simulate long term hydrological performance (see e.g., Thorndahl and Andersen (2021). Especially cases like the green roofs where the performance indeed depend on the antecedent conditions are interesting to investigate under future climate conditions.*

*The paper is generally well written but more details at the conceptual level could help the overall understanding of the proposed procedure.*

> Thank you for your comments. We will intend to take them into account in order to further improve the current manuscript.

**R3_1**

*I suggest describing more in detail the observational outflow data from the green roofs, the physical details of the green roofs, description of dominant processes of the roofs (e.g., in the introduction and not only in the method section), etc.*

> The current introduction includes:
>
> - the hydrological performance of green infrastructure and input data needed for modelling.
> - available observed data in Norway and France.
> - types of downscaling
> - popular temporal downscaling methods.
>
> With respect to this comment (R3_1) and R1_1 we will add a paragraph about green infrastructure and green roofs. With respect to R1_16 we will detail the different behaviour of the green roofs used in this study in terms of dominant process (i.e., the extensive green roof and the detention based extensive green roof).

**R3_2**

*Table 3 shows the differences in retention fraction between observed and projected conditions. It could be interesting to see a similar table with observed values versus modelled performance for the current climate conditions in different MRC modes. Unlike fig 5 and 6, which are difficult to interpret, a table summarizing the performances would clarify this part.*

> The table 3 do not include the different MRC mode in the current system because they would lead to similar performance since retention is often not affected by the downscaling (because of the characteristic time for evapotranspiration). We believe that the indicator for green infrastructure performance must include information on climate variation which lead to performance variation. In our case, due to the different climates investigated and the number of MRC modes, a large amount of information has to be displayed which does not make it suitable for a table in text.

> We will further clarify those figures (5 and 6) according to answers to reviewer 1 and 2. In the light of your comments, we will intend to select an indicator summarizing the performance such as a statistical distance between discharge distribution based on observed and downscaled timeseries. and add it to table 3.

**R3_3**

*Is it possible to summarize the change of climate in some specific parameters, e.g., changes in annual and seasonal precipitation, change in consecutive dry days, temperature, etc. for the different locations? This would*

*help understand the differences between locations and maybe lead to an interpretation of the most important processes for the green roof performance and how the processes change under a change in e.g., temperature, rainfall patterns, etc.*

It would indeed be interesting to clarify the input data for different location and climate period. The data about input are currently detailed in Table 1 The Köppen Geiger classification gives some information on the properties of the climate but we agree that relevant indicators in terms of hydrological performance would be more interesting to share. This table will be extended with some indicators to include shift in climate and represent difference between locations.

---

## Author Response (AR1)

**The changes have been made according to the Author reply to the reviewer comments, namely:**

https://doi.org/10.5194/hess-2021-381-AC1

https://doi.org/10.5194/hess-2021-381-AC2

https://doi.org/10.5194/hess-2021-381-AC3

**Full details available in the file with track change.**

More specifically:

- Typo were checked, abbreviations were reviewed, and significant part of the manuscript was clarified.
- A table of abbreviation have been added.
- The difference between micro canonical and canonical multiplicative random cascade was clarified.
- A paragraph about green infrastructure and green roofs have been added. It discusses the relevance of using the RCP 8.5 for future performance of green infrastructures.
- Details about data from Marseille were added.
- The table was improved to add the main characteristics of each location.
- The part 2.2.2 "Downscaling process" was clarified.
- Notation in equation and figure were unified.
- The part Downscaling models conceptualization and calibration was added. The different models are described and the calibration procedure for the different generator is specified.
- The structure of the different models is available in appendix A.
- An indicator relative Kolmogorov Smirnov, relevant for log survival distribution was developed and used to evaluate the resulting survival distributions.
- A part of the green roof model analysis was moved to the method section.
- A subsection in method about the design perspectives was added.
- In analysis of climate properties, link to other location and other time-scales was added. a figure was added in appendix B to support this analysis.
- The analysis of the downscaled time-series was improved with the relative Kolmogorov Smirnov.
- A figure about autocorrelation for an in-depth analysis was added to show the link between lag-1 autocorrelation depending on time-scale and autocorrelation with multiple lag time for different time-scales.
- Details in design perspectives and the behaviour of the different roofs was added.
- The acknowledgement section was completed.

---

## Author Response (AR2)

[EDITOR]: Dear authors,

Two referees have reviewed the revised manuscript, they were satisfied with the way comments have been addressed. Only minor editing suggestions remain, I invite you to have a look at these and revise the manuscript accordingly.

Additionally, please have a look at the following references, as they seem to be relevant to your work: Versini et al. 2016 (doi: 10.2166/wst.2016.310); Gires et al. 2020 (https://doi.org/10.1080/02626667.2020.1736297)

The revised manuscript will be subject to editor review only.

We thank the editorial team for their support during the reviewing process. The reply to the reviewer comments can be find below. Regarding the suggestion of the editor, Gires et al. 2020 (https://doi.org/10.1080/02626667.2020.1736297) was found very relevant for the current study and therefore added to the current mansucript. Regarding Versini et al. 2016 (doi: 10.2166/wst.2016.310), it was considered by the authors slightly out of the scope of the current paper, the study looks at catchment scale implementation of green infrastructure which lead to different challenges than the modelling of green infrastructure performance at site scale (in the current manuscript), even if the two fields are closely linked. Please note that in accordance with the reviewer comments, several references have been added to strengthen the scientific basis of the study.

Please note that a typo in a formula in the appendix has been corrected, it can be found in the track change. Several other typos or reference issue have been fixed.

[REVIEWER #2]: I was reviewer #2 of the initial submission of the manuscript. Most of my comments have been addressed and I believe that the paper can now be published after few minor changes that do not require a re-review.
(I am using line numbers of the manuscript without track change)

- It should be clearly stated form the abstract that parameters are tuned according to the cascade step, meaning that the underlying idea of scale invariance associated to (multi)-fractal fields is not respected.

It has been clarified.

[REVIEWER #2]: - If parameters change with cascade step; how can you be sure that the ones for the small scales in the future (which are unknown) will remain relevant ? This should be discussed and mentioned.

That is the main hypothesis underlying behind the use of Temperature as a predictor. Since the some precipitation pattern are correlated to some temperature range, the shift in temperature is likely to lead to a shift in parrten which suggest that temperature based MRC have higher robustness. It is also found in this paper that, in most of the location, it did not lead to a significant difference wether temperature is used or not as a preditor. This was briefly discussed L 280, and has been improved according to the reviewer comment. This was clarified in section 2.2.3 and further discussed in 3.2

[REVIEWER #2]: - There are few typos that should be corrected as for example l.31 (therefor → therefore) or l. 117 (consists → Consist), or l. 422 Eq ?? (the equation number is missing)

This has been edited accordingly.

[REVIEWER #2]: - l. 31-32 : it not only a question of computational power, but also of parametrization of the physical process which will still be needed since it will not be possible to go down to kolmogorov scale anyway in the near future.

That remark is very relevant and has therefore been added to the manuscript together with a reference to support that point (https://doi.org/10.1002/qj.2640).

[REVIEWER #2]: - Introduction : some historical perspectives on MRC are missing. Reminding earlier references would be appreciated in terms of scientific context.

Thank you for that comment, those aspects has been added together with several reference (e.g. : Schertzer and Lovejoy, (1987) Gupta and Waymire (1993) Olsson (1998))

[REVIEWER #2]: - l. 80-85 : please explain where does this specific scenario stands among the others ?

RCP8.5 was selected because it leads to a greater shift in temperature. Therefore, the effect of Temperature as a predictor would be more noticeable. The specificity of that scenario regarding climate forcing and gas-emissions has been added.

[REVIEWER #2]: - Eq. 1 and 2 : please add the range of possible values for i and j

"j" stands for the time-scale so is between 1 and 750. The typical notation refers to the cascade level (k, see, e.g.: https://doi.org/10.5194/npg-17-697-2010) but in the case of the current model the time-scale is preferred since it is possible to downscale from hourly as well as from daily which would make the notation ambiguous.

"i" is the time-step, it is conditioned by the size of the original time-series and the time-scale "2j", if j = 1440 min and a dataset of 10000 days, then "i" is in [0, 9999]. This has been clarified.

[REVIEWER #2]: - l. 124-125 : what is the meaning of the sentence "The generators of the MRC models were all time-scale continuous. In practice it means that there is a single set of parameter per generator and not a set per disaggregation step which ensured a parsimonious number of parameters compared to other recent works" ? Please clarify. It may be worth mentioning that other processes truly scale invariant (i.e. no tuning at each step of the distribution) rely on much less parameters as for example the Universal Multifractals (only 2 parameters). Please clarify number of parameters per cascade steps as well.

In the current approach the data analysis lead to observed statistics that depends on time-scale. The microcanonical MRC model used, similarly to Rupp et al. (2009) (doi:10.1029/2008WR007321) a direct analytical formula was found to include the time-scale dependency. This lead to the use of a single set of parameter for the entire process: the formula allow to rescale the distributions used with respect to the time-scale. This sentence has been accordingly reformulated.

The number of parameters per cascade scale is not relevant given the clarification above, but one could argue that given a time-scale, 1 parameter corresponds to the SEP generator (probability of permutation), 1 to 3 parameters for the Zero generator (probability to have a weight equal to zero) and one parameter for the non-Zero generator (sigma). for 8 level it would lead to between 24 and 40 parameters which might be seen as close to the final number of parameters in the parameter set but does not have the same regularity and is subject to more uncertainty.

[REVIEWER #2]: - 2.2.3 : Please provide more explanations on the reasoning that lead to the choice of calibration process ?

The calibration was driven by the dataset analysis in the context of model development. A lot of data proceeding was necessary, it makes the calibration process intuitive from the data analysis perspective but also subject to significant possible improvement in case of model distribution.

> [REVIEWER #2]: - Section 3.1 and Fig. 2 : since as mentioned in answer to referees "observed" is actually "simulated with observed rainfall", it should be clarified what you are calling model validation without actual discharge measurement if I am correct.

For the green roof testing, the discharged is observed/measured. What was mentioned in answer to referees was about the testing of the MRC models. It has been clarified in the methods.

---

## Author Response (AR3)

Dear Editors,

The authors would like to thank the reviewers for their support during the editing process.

[EDITOR] Thank you for addressing the remaining reviewer comments, the clarifications are adequate and the paper is nearly ready to be accepted for publication.

A final technical comment remains to improve readability of the figures: please check the remark that has been upon file validation with respect to the color usage. Additionally, consider using fewer grid lines in several the figures: grid lines density makes them hard to read (esp figures 4, 6, 7, 8).

Figure 4, 6, 7 and 8 (as well as C1 and C2) have been edited to make them easier to read. We decided to keep the grid density but made the colour of the grid lighter so that it is still possible for the reader to use the grid (which is helpful in a log axis context) but does not overwhelm the figure. We are open for more change if the editor finds it necessary.

Regarding colourblind friendly figures:

The colour palette for fig 4 and 7 is: #1b9e77, #d95f02, #7570b3 which is colourblind friendly based on colorbrewer 2.0

The colour palette of fig 6, 8, 9, C1 and C2 is colourblind friendly as it can be seen on:
https://davidmathlogic.com/colorblind/#%231F77B4-%232CA02C-%238C564B

The colour palette for figure 3 and B1, is colourblind friendly in case of 2 colour subplots, or at least differentiated with markers when it is not possible. It should be noted that rectangles directly highlight the information of interest to help the reader.

Figure 1 and 2 do not need colourblind friendly palette dure to their conceptualization.

The author hope that those adjustment will satisfy the editors but are open to change if it is necessary.